# Diffusion Instruction Tuning

**Chen Jin** [1]  **Ryutaro Tanno** [2]  **Amrutha Saseendran** [1]  **Tom Diethe** [1]  **Philip Teare** [1]

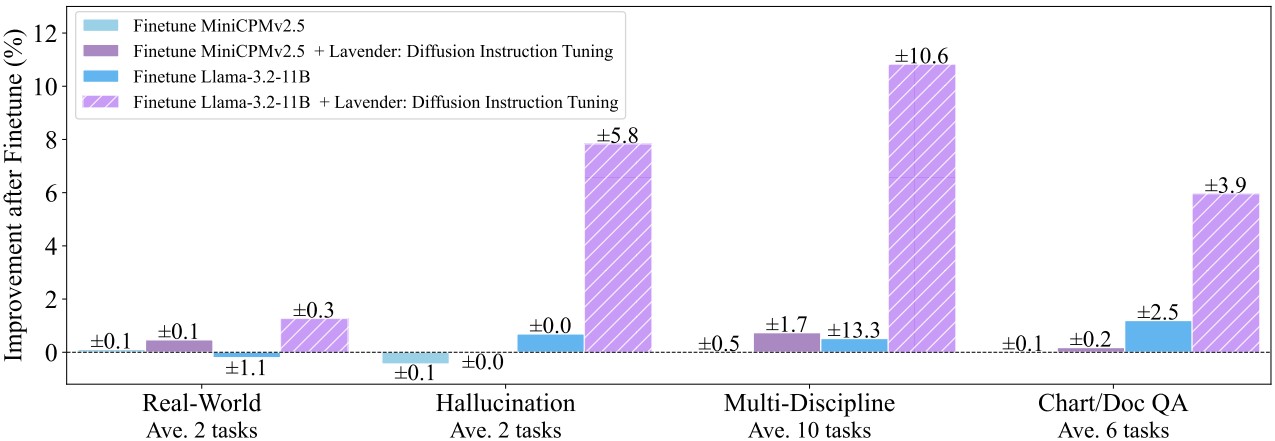

*Figure 1.* Average Performance on 20 Vision-Language Reasoning Benchmarks (Grouped into 4 Categories).

## Abstract

We introduce *Lavender*, a simple supervised fine-tuning (SFT) method that boosts the performance of advanced vision-language models (VLMs) by leveraging state-of-the-art image generation models such as Stable Diffusion. Specifically, Lavender aligns the text-vision attention in the VLM transformer with the equivalent used by Stable Diffusion during SFT, instead of adapting separate encoders. This alignment enriches the model's visual understanding and significantly boosts performance across in- and out-of-distribution tasks. Lavender requires just 0.13 million training examples—2.5% of typical large-scale SFT datasets—and fine-tunes on standard hardware (8 GPUs) in a single day. It consistently improves state-of-the-art open-source multimodal LLMs (e.g., Llama-3.2-11B, MiniCPM-Llama3-v2.5), achieving up to 30% gains and a 68% boost on challenging out-of-distribution medical QA tasks. By efficiently transferring the visual expertise of image generators with minimal supervision, Lavender offers a scalable solution for more accu-

rate vision-language systems. Code, training data, and models are available on the project page.

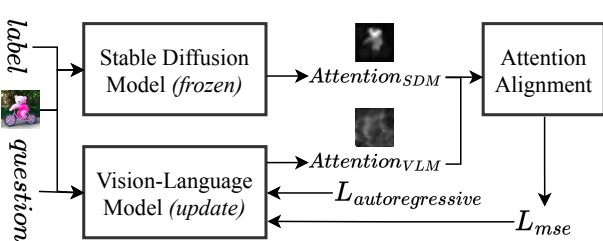

*Figure 2.* ***Lavender*: Diffusion Instruction Tuning.** Lavender uses the text-vision attention maps of a Stable Diffusion Model, $Attention_{SDM}$, as a guiding objective for the attention of the target vision-language model (VLM), $Attention_{VLM}$. The *Attention Alignment* module employs a 3-Layer ConvNet to transform $Attention_{VLM}$ to match $Attention_{SDM}$ via an MSE loss, acting as a regularisation term during supervised fine-tuning.

## 1. Introduction

Training frontier foundation models from scratch costs millions of dollars at minimum, requiring hundreds of GPUs and millions to billions of data (DeepSeek-AI et al., 2024). This challenge is even more pronounced in multimodal settings: vision-language models (VLMs) often face data scarcity because collecting paired image-text datasets is expensive (Zhu et al., 2024). A common workaround is to

[1]Centre for AI, AstraZeneca, Cambridge, UK [2]Google DeepMind, UK. Correspondence to: <chen.jin@astrazeneca.com>.

*Proceedings of the $42^{nd}$ International Conference on Machine Learning*, Vancouver, Canada. PMLR 267, 2025. Copyright 2025 by the author(s).

apply supervised fine-tuning (SFT) on a pretrained large language model (LLM), leveraging its abundant text-only pre-training and adjusting bridging layers or additional encoders with limited image-text pairs (Liu et al., 2024c; Covert et al., 2024; Jiang et al., 2023). However, these methods typically overlook the importance of transformer-level attention alignment within the LLM core—a key component for effectively expanding text-based models into the visual domain.

Precise visual-text alignment is crucial for advanced multi-modal reasoning. While both VLMs and diffusion models (DMs) process text and images, they diverge in their generation objectives. We observe that DMs, such as Stable Diffusion (Rombach et al., 2021), which reconstructs images at the pixel level, appear to learn more precise text-vision attention maps than VLMs that are optimised solely for text token generation (Figure 3).

In this work, we demonstrate that the high-quality cross-attention maps from these DMs indeed offer a useful target for guiding the text-vision attention in VLMs during SFT, thus improving word-to-region alignment and the overall performance. We introduce **Lavender** (**L**anguage-**a**nd-**V**ision fine-tu**n**ing with **D**iffusion Align**er**), *the first framework to directly align VLM transformer attention layers with those of Stable Diffusion* (Figure 1(c)). Specifically, during SFT, Lavender transfers diffusion-based attention distributions to VLMs, enhancing core visual-textual interactions. To mitigate catastrophic forgetting, we additionally propose several attention aggregation methods and training strategies that preserve existing VLM competencies.

We begin by verifying Lavender on a small OpenFlamingo model: entropy and visual analyses show Lavender aligns VLM attention with DM attention. Leveraging Stable Diffusion to *offline* extract per-word attention on 130k label-image pairs-*no extra training cost*-Lavender yields notable gains over autoregressive finetuning on 20 diverse benchmarks, including up to 70% improvement on OpenFlamingo across seven benchmarks. For Llama 3.2-11B, fine-tuned on in- and out-of-distribution data, performance improves by up to 30% on 19 benchmarks, surpassing comparable small open-source models by 50%. On self-attention-only MiniCPMv2.5, it achieves up to 4% gains.

This advantage extends to severely OOD domains, evidenced by a 68% boost on the WorldMedQA medical benchmark for Llama 3.2-11B. Further analyses reveal that larger fine-tuning sets help Lavender resist overfitting more effectively than autoregressive baselines, and the aligned attention maps yield finer-grained visual understanding. Together with qualitative evidence of improved VLM attention, these results confirm Lavender's premise: *diffusion-based attention distributions effectively align visual and textual representations*, fostering more robust, data-efficient VLMs.

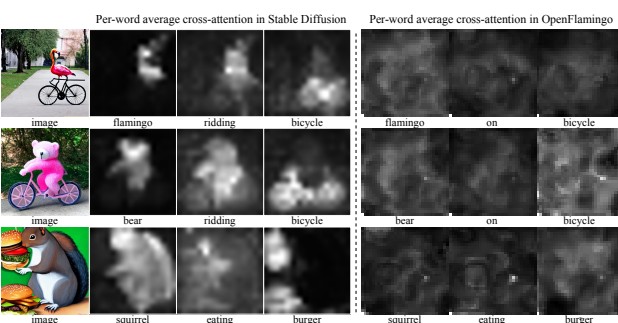

*Figure 3.* **Image generation models (Stable Diffusion on the left) exhibit stronger word-to-region attention alignment than VLMs (Open-Flamingo on the right).** Per-word average attention maps suggest that diffusion models may be closer to an ideal distribution correlating image regions with textual tokens.

Ablation studies reveal that the method of attention aggregation and the choice of layers for fine-tuning are critical to performance. Learned aggregation strategies outperform manual ones and lightweight pretraining of an Aligner Network helps prevent catastrophic forgetting on small datasets. LoRA fine-tuning delivers faster improvements, full fine-tuning proves more effective for handling complex tasks. Aligning all cross-attention layers proves most effective, highlighting the importance of precise attention alignment.

**In summary,** we introduce *Lavender*, a novel framework that transfers "visual expertise" from text-to-image diffusion models to vision-language models *without additional annotations*. By aligning attention distributions, Lavender enhances word-to-region grounding, improves fine-tuning efficiency, and boosts model robustness, particularly in out-of-distribution settings. Moreover, our architecture-agnostic attention alignment loss is compatible with RL post-training, offering scalable diffusion-guided feedback instead of costly, subjective human vision feedback. Beyond addressing data scarcity, Lavender demonstrates that pretrained generative models can guide multimodal learning in a scalable and compute-efficient manner. This approach bridges two expert paradigms—language and vision generation—into a more unified, capable system. Our findings suggest broader applications in multimodal AI, offering a modular and privacy-friendly alternative to closed-source models. We open-source our work to encourage further exploration of diffusion-guided alignment, unlocking new possibilities in vision-language reasoning.

## 2. Diffusion Instruction Tuning

We aim to enhance a pretrained Vision-Language Model (VLM) by leveraging attention distributions from a pretrained Diffusion Model (DM). We assume there is an *ideal* attention distribution that maximises VLM performance and that the DM's attention is closer to this ideal distribution.

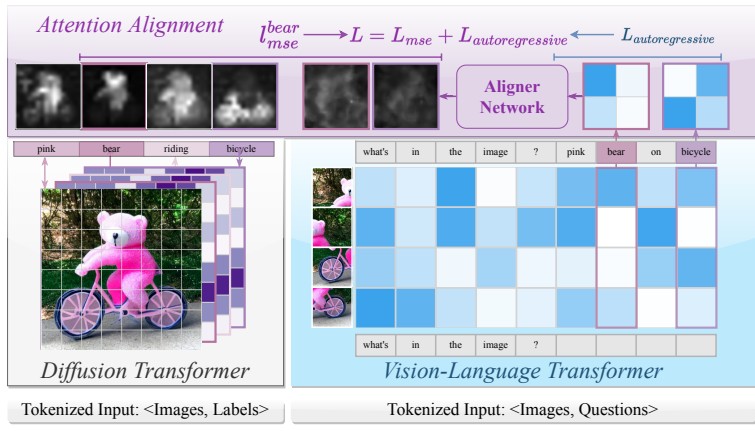

**Algorithm 1** Diffusion Instruction Tuning

**Require:** $D = \{(x^{(i)}, y^{(i)})\}, \theta_D, \theta$, scale $\lambda$
**Ensure:** Fine-tuned VLM parameters $\theta$
    **Stage 1: Precompute DM Attention (run once)**
    **for** each $(x^{(i)}, y^{(i)}) \in D$ **do**
        $A_{\text{DM}}^{(i)} \leftarrow p_{\text{DM}}(a \mid x^{(i)}, y^{(i)}; \theta_D)$
    **end for**
    **Stage 2: Fine-Tune VLM**
    **repeat**
        Sample batch $B \subseteq D$, set $L_{\text{VLM}}(\theta) = 0, L_{\text{att}}(\theta) = 0$
        **for** each $(x^{(i)}, y^{(i)}) \in B$ **do**
            Compute $p_{\text{VLM}}(a \mid x^{(i)}, y^{(i)}; \theta)$
            $\delta^{(i)}(\theta) \leftarrow \text{Aligner}(p_{\text{VLM}}(a)) - A_{\text{DM}}^{(i)}$
            $L_{\text{VLM}}(\theta) += - \log p(y_l^{(i)} \mid x^{(i)}, y_q^{(i)}; \theta)$
            $L_{\text{att}}(\theta) += \|\delta^{(i)}(\theta)\|^2$
        **end for**
        $L_{\text{total}}(\theta) \leftarrow L_{\text{VLM}}(\theta) + \lambda L_{\text{att}}(\theta)$
        Update $\theta \leftarrow \theta - \eta \nabla_\theta L_{\text{total}}(\theta)$
    **until** convergence

*Figure 4.* Sketch of Diffusion Instruction Tuning (left) and a short pseudo code (right), whose full version is available in Appendix F.

## 2.1. Models and Notation

**Vision-Language Model (VLM).** Let $\theta$ be the VLM parameters, pretrained for tasks such as image captioning or question-answering. It models:

$$p(y_l | x, y_q; \theta), \tag{1}$$

where $x, y_q, y_l$, as image, question, and label answer.

**Diffusion Model (DM).** Let $\theta_D$ be the DM parameters, which remain fixed during our procedure. It models:

$$p(x | y; \theta_D). \tag{2}$$

**Attention Distributions.** We write:

$$p_{\text{VLM}}(a | x, y; \theta), \quad p_{\text{DM}}(a | x, y; \theta_D), \tag{3}$$

We hypothesise that $p_{\text{DM}}(a | x, y; \theta_D)$ is closer to the optimal posterior attention distribution $p^*(a | x, y)$ than $p_{\text{VLM}}(a | x, y; \theta)$, and the two can be aligned by projecting $p_{\text{VLM}}$ into a comparable space using small learnable layers.

## 2.2. Assumptions

*Ideal attention in Vision-Centric Tasks:* An attention distribution $p^*(a | x, y)$ minimises the next-token prediction loss of VLM, $L_{VLM}$; *DM Attention Proximity:* Empirically, the DM's attention is more concentrated (lower entropy) and hence closer to $p^*$ than the VLM's, supported by Figure 3, experiments in Section 6.1 and detailed justifications in Appendix G; *Shared Dataset:* Both models use the same image-text set $\{(x^{(i)}, y^{(i)})\}$; *Fixed DM Parameters:* $\theta_D$ is kept fixed; only $\theta$ is updated; *Pretrained VLM Parameters:* $\theta$ is further fine-tuned with an attention alignment loss.

## 2.3. Bayesian Derivation

Our objective is to update the VLM parameters $\theta$ such that the model not only performs well on its primary task but also aligns its attention mechanism with that of the DM. We formalise this objective within a Bayesian framework.

**Posterior Distribution:** We aim to find the posterior distribution of the VLM parameters given the data $D$ and the DM's attention distributions:

$$p(\theta | D, A_{\text{DM}}) \propto p(D | \theta) \, p(A_{\text{DM}} | \theta) \, p(\theta), \tag{4}$$

where $A_{\text{DM}} = \{p_{\text{DM}}(a | x^{(i)}, y^{(i)}; \theta_D)\}$ is the collection of attention outputs derived from the DM's conditional distribution, and $p(\theta)$ is the prior over the VLM parameters.

**Likelihood of the Data:** The likelihood of the data given $\theta$ is:

$$p(D | \theta) = \prod_i p(y_l^{(i)} | x^{(i)}, y_q^{(i)}; \theta). \tag{5}$$

The negative log-likelihood corresponds to the standard loss function $L_{\text{VLM}}(\theta)$ used to fine-tune the VLM:

$$L_{\text{VLM}}(\theta) = - \sum_i \log p(y_l^{(i)} | x^{(i)}, y_q^{(i)}; \theta). \tag{6}$$

**Likelihood of the DM's Attention:** We model the likelihood of observing the DM's attention given the VLM's parameters, denoted as $p(A_{\text{DM}} | \theta)$. To simplify the notation and make the equations more concise, we introduce:

$$\delta^{(i)}(\theta) = p_{\text{VLM}}(a | x^{(i)}, y^{(i)}; \theta) - p_{\text{DM}}(a | x^{(i)}, y^{(i)}; \theta_D). \tag{7}$$

This represents the pointwise difference between the VLM's and DM's attention distributions for the $i$-th data point, serving as a measure of divergence at each attention location $a$. Assuming that these differences are Gaussian-distributed with equal variance, the likelihood can be expressed as:

$$p(A_{\text{DM}} | \theta) \propto \exp\left(-\frac{\lambda}{2} \sum_i \left\|\delta^{(i)}(\theta)\right\|^2\right). \tag{8}$$

This corresponds to the attention alignment loss $L_{\text{att}}(\theta)$:

$$L_{\text{att}}(\theta) = \sum_i \left\| \delta^{(i)}(\theta) \right\|^2. \tag{9}$$

By minimizing $L_{\text{att}}(\theta)$, i.e., the MSE loss, we encourage the VLM's attention to align with that of the DM, guiding it toward the optimal posterior attention distribution $p^*(a|x, y)$.

For simplicity, we assume a non-informative prior $p(\theta)$. Consequently, the posterior distribution in the previous equation Equation (4) is governed primarily by $p(D|\theta)$ and $p(A_{\text{DM}}|\theta)$. If regularisation is needed, a more informative prior can be seamlessly incorporated. Combining the terms, the negative log-posterior (up to a constant) becomes:

$$L_{\text{total}}(\theta) = L_{\text{VLM}}(\theta) + \lambda L_{\text{att}}(\theta). \tag{10}$$

Here, $\lambda$ balances the importance of aligning the attention distributions with the primary task. We fully justify the inclusion of the attention alignment loss in Appendix H.

### 2.4. Practical Implementation

For each sample $(x^{(i)}, y^{(i)})$, we extract per-word attention from both models. Then, we fine-tune $\theta$ by minimizing:

$$L_{\text{total}}(\theta) = -\sum_i \log p\big(y_l^{(i)}|x^{(i)}, y_q^{(i)}; \theta\big) + \lambda \sum_i \|\delta^{(i)}(\theta)\|^2. \tag{11}$$

This is model-agnostic, requires no additional data, and encourages the VLM's attention to become more focused by following the DM's distributions.

## 3. Attention Alignment

We discuss how to compute per-word attention in VLMs and DMs. Although both employ attention to capture vision-text interplay, their attention aggregation differs (Figure 4). Understanding these distinctions is key to effective alignment.

### 3.1. Attention Aggregation in Diffusion Models

Text-guided diffusion models generate images from textual input by iteratively denoising a random-noise image. During each denoising step, *cross-attention* layers enable the model to focus on relevant textual tokens. Specifically, queries $Q$ are derived from the noisy image $x_t$, while keys $K$ and values $V$ come from the text embedding $v$:

$$Q = f_Q(x_t), \quad K = f_K(v), \quad V = f_V(v), \tag{12}$$

where $f_Q$, $f_K$, and $f_V$ are pretrained projection matrices of DM. The attention map $M$ is then computed as:

$$M = \text{Softmax}\big((QK^\top)/\sqrt{d}\big). \tag{13}$$

Prior work (Hertz et al., 2022) shows that averaging these maps across layers and time steps reveals meaningful correspondences between words and image regions. The resulting per-word attention distributions $p_{\text{DM}}(a|x, y; \theta_D)$ indicate salient image regions for each token, as shown in Figure 3. We leverage these maps as a proxy for the optimal posterior attention distribution $p^*(a|x, y)$, guiding the VLM's alignment toward more focused vision-text interactions.

### 3.2. Attention Aggregation in Vision-Language Models

In VLM transformers, each text token $T_t$ attends to image patch tokens $T_p$ across multiple heads and layers, producing attention weights $w_{(t,p)}^{hl}$, where $h \in H, l \in L, t \in T, p \in P$, with $H, L, T, P$ being heads, layers, tokens, and patches, respectively. These weights capture semantic and spatial relations between tokens and patches. To obtain a single-channel per-word map, we aggregate $N_{\text{text}} \times N_{\text{patch}} \times H \times L$ attention heads/layers into a $(N_{\text{text}} \times N_{\text{patch}})$ matrix. Then we reshape the patch dimension into a $\sqrt{N_{\text{patch}}} \times \sqrt{N_{\text{patch}}}$ grid, approximately reconstructs the original layout of the image, which we verify in Appendix J. This procedure yields interpretable saliency maps—each row corresponds to a text token's focus on the image patches—facilitating alignment with DM attention (see Figure 5).

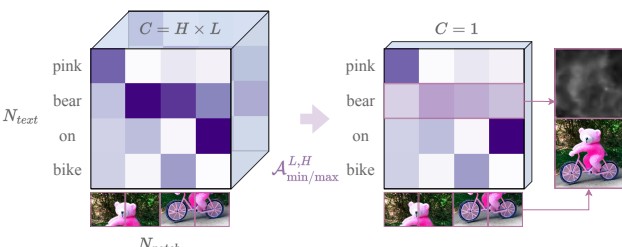

*Figure 5.* **An illustration of the attention aggregation process in VLMs.** Attention weights between text tokens and image patches are aggregated to form per-word saliency maps that approximate the spatial layout of the image.

#### 3.2.1. SIMPLE AGGREGATION FUNCTIONS

A straightforward approach is to pool attention weights $\mathbf{A}$ (i.e., $w_{(t,p)}^{hl}$) across heads $H$ and layers $L$ via mean or max. We consider four strategies:

$$\mathcal{A}_{\text{mean/max}}^{(L,H)}(\mathbf{A}) \in \{\text{max-max, max-mean, mean-max, mean-mean}\},$$

where each denotes a combination of $\{\text{Max}, \text{Mean}\}$ over $H$ and $L$. This yields a single per-word attention map, capturing a coarse measure of word-to-patch alignment.

#### 3.2.2. ATTENTION FLOW

Proposed by Abnar & Zuidema (2020), *attention flow* cumulatively combines multi-layer attention to capture deeper interactions than simple pooling. Starting with the first layer's attention $A^{(1)}$, we iteratively merge subsequent layers $A^{(l)}$ via element-wise multiplication or addition:

$$\bar{A} \leftarrow \bar{A} \circ A^{(l)} \quad \text{or} \quad \bar{A} \leftarrow \bar{A} + A^{(l)}.$$

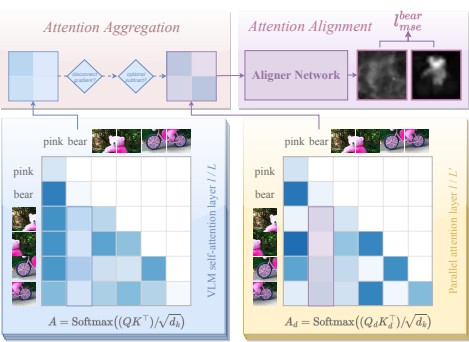

Figure 6. **Learning to aggregate with parallel attention.** The demonstration is based on a self-attention VLM. The parallel attention constitutes about $1/5^{th}$ of the total VLM layers ($L' \in L$).

This method, also used by Lin et al. (2024) at *sentence level*, is extended here to finer-grained *word-level attention maps*, potentially revealing semantic correlations overlooked by simpler aggregation. Further details are in Appendix I.

### 3.2.3. LEARNING THE ATTENTION AGGREGATIONS

Beyond fixed pooling methods, we introduce *parallel* cross-attention parameters ($W_{Q_d}, W_{K_d}, W_{V_d}$) alongside the pre-trained projections ($W_Q, W_K, W_V$). By preserving the original attention mechanisms and learning new ones, we *capture richer semantic correlations without overwriting existing weights*. During each forward pass, we compute both the original attention $A$ and parallel attention $A_d$, then use $A_d$ as the VLM's attention to align with the DM:

$$A = \mathrm{Softmax}\big((QK^\top)/\sqrt{d_k}\big), \qquad (14)$$

$$A_d = \mathrm{Softmax}\big((Q_d K_d^\top)/\sqrt{d_k}\big). \qquad (15)$$

Empirically, learning parallel cross-attention in about $1/5^{th}$ of the layers (see Figure 6) suffices for effective alignment while retaining the original VLM's core knowledge.

### 3.3. Aligner Network

To refine the parallel (or aggregated) attention $A_d$ into a single-channel map comparable to $p_{\mathrm{DM}}(a|x, y; \theta_D)$, we introduce a lightweight Aligner network inspired by Squeeze-and-Excitation (Hu et al., 2018). It contains several small (3-5) layers (MLP or convolution) that expand, apply non-linear activations, then squeeze back to a single-channel map. Empirically, we found *convolutional layers better capture local spatial cues than MLP*, detailed comparisons are provided in Appendix N.1. During fine-tuning, the Aligner output is compared to the DM's attention via:

$$L_{\mathrm{att}}(\theta') = \sum_i \big\| \mathrm{Aligner}\big(A_d^{(i)}\big) - p_{\mathrm{DM}}(a|x^{(i)}, y^{(i)}; \theta_D)\big\|^2, \tag{16}$$

guiding the VLM's attention toward the DM's more focused distribution, capturing complex semantic correlations while preserving the original pretrained parameters.

### 3.4. Lavender Integration

**Cross-Attention.** For VLMs with dedicated cross-attention layers, each head produces word-to-patch weights $w_{(t,p)}^{hl}$ mapping text tokens $T_t$ to image patches $T_p$. We can reshape these weights into spatial grids and aggregate across heads/layers, then apply the Aligner network to yield final per-word saliency maps comparable to DM attention.

**Self-Attention Only.** When both text and image patches are interleaved in a single sequence, tokens attend to each other in a bidirectional or causal manner. To extract word-to-patch correlations, we first separate text tokens from image patches, filter out irrelevant attention connections, and reshape the relevant weights into a grid. Despite the extra steps (e.g., token indexing, handling masks, interpolation), the principle remains the same: aggregate the transformed weights into per-word saliency maps, then align them with DM attention via the Aligner, as demonstrated in Figure 6. This allows Lavender to improve vision-text alignment even in fully self-attentive architectures. A more detailed explanation of both scenarios is provided in Appendix K.

## 4. Implementation (Short Summary)

We integrate Lavender with three Vision-Language Models (VLMs)—cross-attention VLMs (OpenFlamingo, Llama 3.2-11B-Vision Instruct), and self-attention VLMs (MiniCPM-Llama 3-v2.5)—and use Stable Diffusion v1.4 (Rombach et al., 2021) to provide per-word attention targets. We add a lightweight Aligner network and an attention alignment loss to guide the VLM toward the DM's more focused distributions. Further details on DM inversion process, attention extraction, training hyperparameters, and computing environments are provided in Appendix L.

## 5. Training and Dataset (Short Summary)

We adopt several strategies to stabilise alignment objectives and preserve a VLM's pretrained capabilities. First, optionally *pretrain only the Aligner network*, freezing the rest of the model to absorb DM-based attention signals without disrupting existing representations. Second, *attention aggregation* (§3) can stabilise the alignment. The *Aligner* has a configurable depth balancing complexity and accuracy. We also use *PEFT* (e.g., LoRA) to prevent catastrophic forgetting. Finally, we apply 'root word match' or 'exact word match' to select predicted words for alignment. For *datasets preparation*, we process images with Stable Diffusion to obtain per-word attention targets across Flickr30k, Laion50k, RLAIF-V 83k, and OCRVQA30k. Full details, including hyperparameters, dataset preprocessing, and design choices, are provided in Appendix M.

# 6. Experiments and Results

We first validate Lavender on a smaller setup with Open-Flamingo (§6.1) and then scale it to MiniCPMv2.5 (Yao et al., 2024) and Llama 3.2-11B-Vision Instruct (Dubey et al., 2024), evaluating on 20 standard VLM benchmarks and comparing against 23 baseline models (§6.2). We further analyse data overlapping (§6.3), scaling behaviour (§A.2), out-of-distribution medical tests (§6.5), and provide qualitative visualizations in §6.6.

## 6.1. Empirical Verifications

We find that: 1) Figure 7 shows entropy measurements across Flickr30k, RLAIF-V83k, and OCRVQA30k confirming that DM attention distributions are lower in entropy than VLM attention, supporting they are closer to the ideal distribution $p^*(a|x, y)$ in vision-centric tasks. 2) Figure 8 shows how VLM attention maps align with DM maps, achieved by using *exact word match sampling*, *convolutional Aligner networks*, and *moderate learning rates* during Lavender fine-tuning of OpenFlamingo on Flickr30k. 3) This alignment also boosts text generation performance, as evidenced in Figure 9 by improvements in COCO captioning scores correlating with reduced MSE loss when jointly minimising $L_{\text{VLM}}$ and $L_{\text{att}}$. 4) Benchmarking Lavender against autoregressive fine-tuning shows consistent zero-shot gains, with up to 72% improvement (Figure 10). Detailed results and further analysis are discussed in Appendix §N.

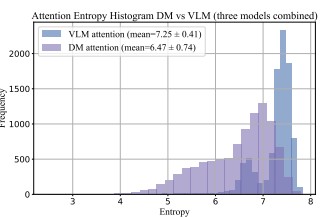

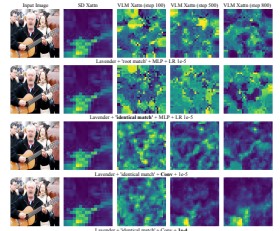

*Figure 7.* **DM attention is more concentrated compared to three VLMs attention (OpenFlamingo, MiniCPM-v2.5, and Llama 3.2-11B).** Full results are available in Appendix N.1.

*Figure 8.* **Aligning VLM attention with DM attention for the word "guitar."** Each row adds a training technique. See Appendix N.1 for more examples.

## 6.2. Scaled Results with Lavender

Next, we scale experiments to MiniCPMv2.5 and Llama 3.2-11B-Vision-Instruct, fine-tuning on RV83k, Flk30k, and OV30k datasets. Models were trained using autoregressive and Lavender methods combined with LoRA or full fine-tuning and evaluated on 20 multimodal benchmarks, comparing against 23 baseline models.

**Evaluation and Baselines.** Lavender was tested across diverse benchmarks grouped by task type (Chart&Doc, perception, real-world understanding, hallucination detection).

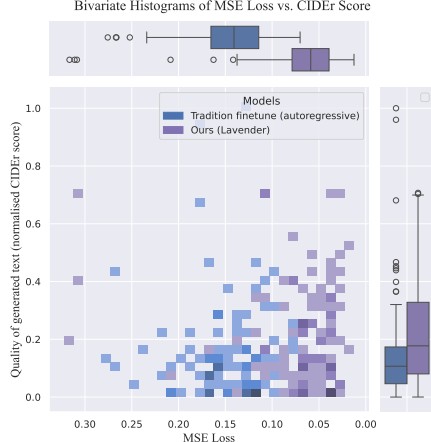

*Figure 9.* **Lower MSE aligns with higher caption quality.** Results on COCO using Lavender-OpenFlamingo F.T. with Flickr30k.

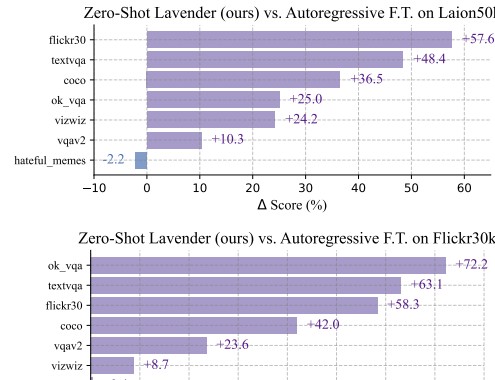

*Figure 10.* **Lavender surpasses the autoregressive baseline by up to 72% in zero-shot evaluations,** with models trained on Laion50k or Flickr30k. The negative gain on Hateful Memes stems from its unique ranking task rather than captioning.

Baselines include three groups: small budget-constrained models ($< 10$B parameters), small data-heavy models ($< 20$B), and large-scale SoTA models ($> 20$B). Details of benchmarks, baselines, and evaluation are in Appendix O.

**Main Results with MiniCPM-V-2.5 and Llama 3.2-11B.** Figure 11 compares results for MiniCPM-V-2.5 (self-attention-only) and Llama 3.2-11B-Vision-Instruct (with cross-attention). For MiniCPM-V-2.5, Lavender outperforms autoregressive fine-tuning on 16/20 tasks, improving performance by up to 4% while limiting drops to -1%, despite challenges posed by dataset reuse and the self-attention mechanism. For Llama 3.2-11B, Lavender achieves up to 30% gains on 19/20 benchmarks with LoRA and up to 25% on 17/20 with full fine-tuning. These results underscore Lavender's robustness across VLMs, with our experiments indicating that explicit cross-attention modules align more effectively than self-attention-only models like MiniCPM.

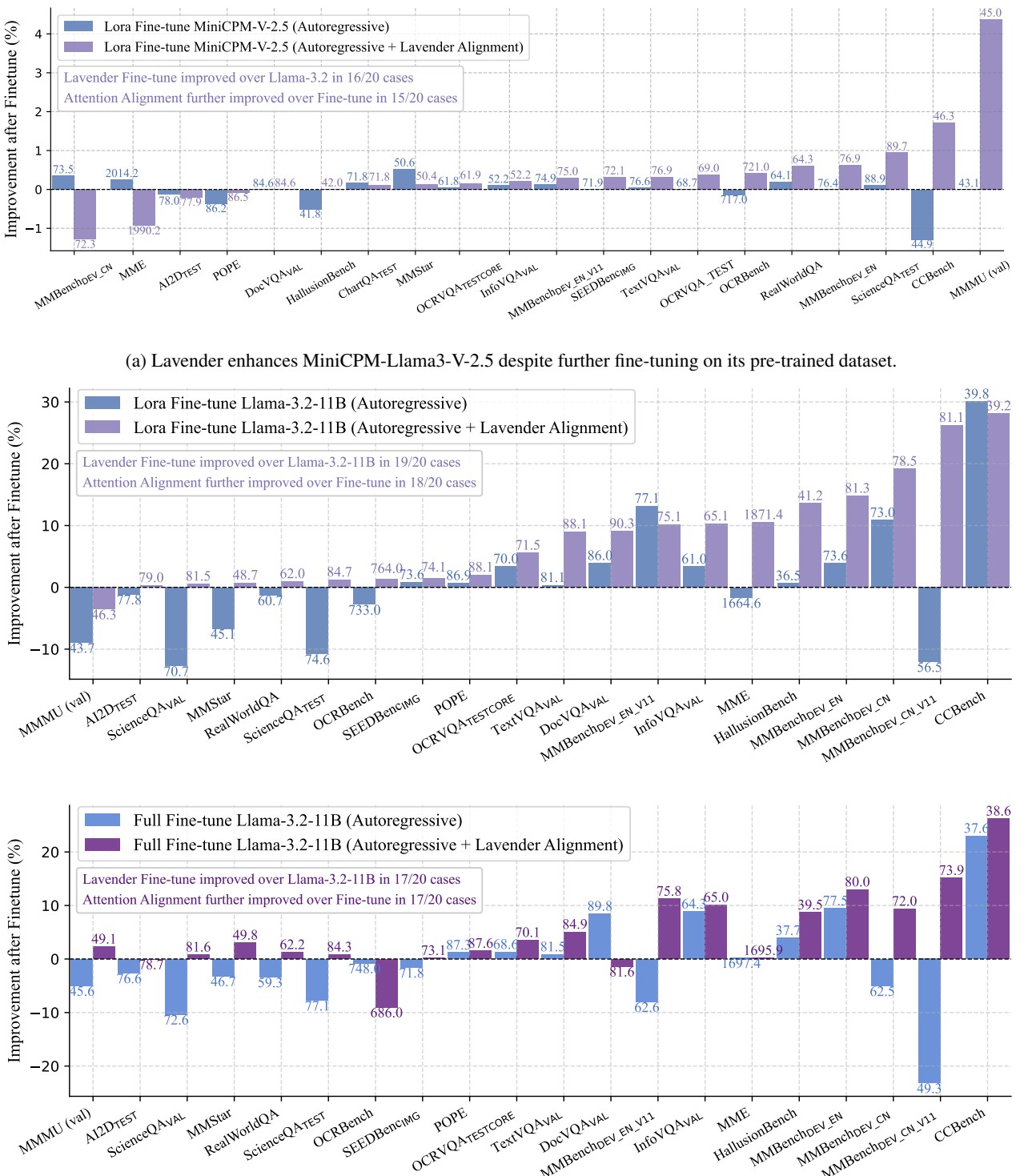

(a) Lavender enhances MiniCPM-Llama3-V-2.5 despite further fine-tuning on its pre-trained dataset.

(b) Lavender mitigates catastrophic forgetting for Llama-3.2-11B fine-tuned on a new dataset.

*Figure 11.* **Relative Improvement Over Baseline on 20 Benchmarks.** (a) Results for MiniCPM-Llama3-V-2.5 fine-tuned on its original RV83k dataset. Lavender improves performance on 16/20 benchmarks with gains up to 4%, while limiting performance drops to -1%, primarily on an out-of-distribution Chinese benchmark. (b) Results for Llama-3.2-11B-Vision-Instruct fine-tuned on a mixture of RV83k, Flk30k, and OV30k using LoRA and full fine-tuning strategies. Lavender outperforms autoregressive fine-tuning on 18/20 (LoRA) and 17/20 (Full-FT) benchmarks, achieving up to 30% and 25% improvement, respectively, while mitigating catastrophic forgetting. Across both models, Lavender demonstrates consistent benefits over autoregressive fine-tuning, particularly in reducing catastrophic forgetting.

**Benchmarking Lavender Against External Baselines.**
Lavender focuses on enhancing VLM performance beyond standard next-token fine-tuning. To contextualise its improvements, we compare Lavender against a range of baseline models across 16 benchmarks. Key results are summarised in Figure 15, with detailed comparisons in Appendix Table 1. For *Small Budget-Constrained Models*, Lavender achieves up to 50% improvement, with minor deficits (within 4%) on benchmarks like SEED-IMG. Among *Small Data-Heavy SOTA Models*, Lavender outperforms autoregressive baselines, despite using significantly smaller datasets (38x–384x less data), with gaps attributed to dataset differences (Section 6.3). For *Large SOTA Models*, Lavender performs comparably to some closed-source models on certain tasks demonstrated in Figure 12.

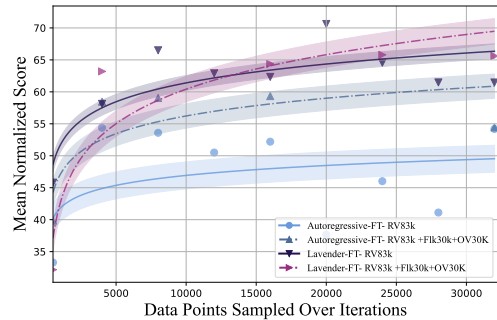

*Figure 14.* **Lavender scales better and mitigates overfitting compared to autoregressive fine-tuning, with larger datasets reducing variability.** This plot shows the mean normalised performance across eight benchmarks for two dataset configurations after fine-tuning Llama 3.2-11B. Markers indicate observed performance with trendlines, and shaded regions show variability as 1 standard deviation around the mean. Full results with four dataset configurations are in Appendix Figure 18 and Figure 28.

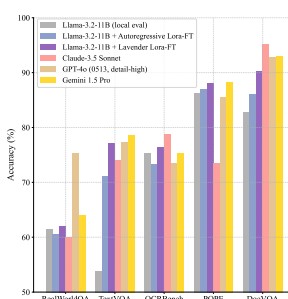
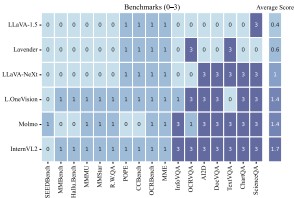

*Figure 12.* **Lavender matches certain high-resource SOTA models.**

*Figure 13.* **Minimal data overlap supports Lavender's generalisability.** Overlap scores are based on: 3 for explicitly shared benchmark datasets, 1 for shared sources (e.g., COCO), and 0 for no overlap.

### 6.3. Data Overlap Analysis

To contextualise benchmark results, we analyse the overlap between fine-tuning and benchmark datasets. Using a qualitative approach (Figure 13), we find that fine-tuning datasets for some SoTA models show higher overlap with benchmarks. Lavender's fine-tuning dataset exhibits low overlap, similar to LLaVA-1.5 (Liu et al., 2024a), highlighting its strong generalisability. This analysis focuses on fine-tuning, where smaller datasets pose higher overfitting risks, though pretraining overlap can't be ruled out.

### 6.4. Scaling Behaviour

Lavender, a model-agnostic approach, is tested on small fine-tuning datasets in this work due to computational constraints. To evaluate its scalability, we fine-tune Llama-3.2-11B on combinations of RV83k, Flk30k, and OV30k using autoregressive and Lavender methods with LoRA or full fine-tuning. Results in Figure 14 show average performance across eight benchmarks. The findings indicate that Lavender scales better with increased data, effectively reducing overfitting—a common issue with autoregressive fine-tuning on small datasets.

### 6.5. Severely Out-of-Distribution Medical Benchmark

We evaluate model generalisation on the extreme OOD WorldMedQA-V (Duan et al., 2024), a multilingual, multimodal medical VQA dataset. It includes 568 multiple-choice questions in four low-resource languages, paired with medical images. The dataset's focus on medical exams and rare languages makes it ideal for testing generalisation beyond typical fine-tuning domains. Figure 16 shows that Lavender improves Llama-3.2-11B's performance by 68%, surpassing six open-source models (6B–34B) and narrowing the gap with large closed-source models from 43% to 10%.

### 6.6. Qualitative Results with Llama 3.2-11B

Lavender effectively aligns VLM attention maps with those of DMs and enhances performance on VQA tasks. Examples of aligned attention maps (Figure 19) show that Lavender's attention maps correlate well with semantic regions similar to those of DMs. Additionally, Figure 17 highlights improved performance on diverse VQA benchmarks attributed to the better localisation and interpretation of visual elements with Lavender. Full details in Appendix A.1.

## 7. Ablation and Analysis Summary

We conducted extensive ablation studies to assess the key components of Lavender and their impact on performance. Key findings include: 1) *Attention Aggregation*: Learned aggregation consistently outperformed manual methods like attention flow due to its adaptability and scalability (Figure 20, Figure 21). 2) *Pretraining the Aligner*: Pretraining the Aligner Network significantly mitigated catastrophic forgetting compared to plain fine-tuning, with longer pretraining benefiting more complex benchmarks (Figure 22). 3) *Fine-tuning Strategy*: LoRA offered better short-term results,

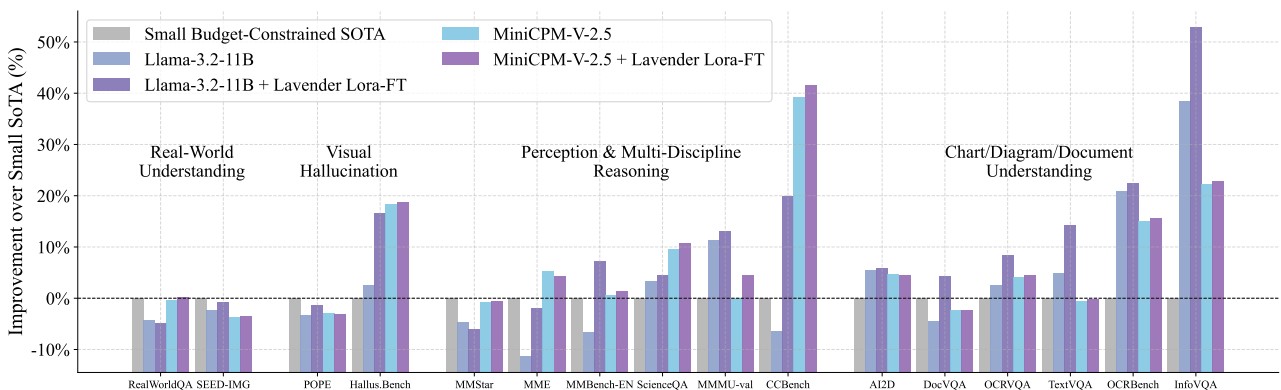

Figure 15. **Lavender improves MiniCPM-V-2.5 and Llama-3.2-11B, surpassing Small Budget-Constrained SOTA by up to 50%.** Key zero-shot accuracy results across 16 VLM benchmarks, with the greatest gains in *Chart, Diagram, and Document Understanding*. Moderate improvements are seen in *Perception* and *Hallucination*, while *Real-World Visual Understanding* shows the smallest gains. Detailed results including 23 baseline models are presented in Appendix Table 1.

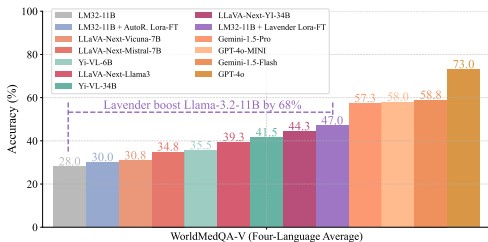

Figure 16. **Lavender boosts Llama-3.2-11B's performance on the OOD WorldMedQA benchmark by 68%.** Results are based on fine-tuning Llama-3.2-11B-Vision-Instruct on RV83k, Flk30k, and OV30k using autoregressive or Lavender methods with LoRA. Accuracy reflects average performance across four low-resource medical VQA subsets, evaluated via Vlmevalkit (Duan et al., 2024). Example results in Appendix Table 2.

while full fine-tuning showed advantages on more complex tasks, suggesting complementary benefits (Figure 11). 4) *Layer Alignment*: Aligning all eight cross-attention layers in Llama-3.2-11B was most effective, outperforming partial alignments (Figure 23). Full details and additional analyses are provided in Appendix B.

## 8. Failure Strategies and Limitations

**Failure Strategies.** Despite Lavender's overall success, certain approaches proved less effective. *Fully fine-tuning without pretraining* often destabilised the model, particularly on small datasets, though LoRA offered a more robust alternative (Figure 22). F*requent switching between training strategies* (e.g., alternating between full fine-tuning and aligning cross-attention subsets) harmed performance due to the model's inability to stabilise objectives during short training periods. Additionally, *incorporating extra datasets*, such as OCRVQA, sometimes reduced performance (Figure 14), suggesting risks of overfitting with specific data.

Full details are provided in the Appendix C.

**Limitations and Future Works.** Lavender's evaluation was limited to datasets of up to 0.13M samples, far smaller than those used by state-of-the-art models, which scale up to 50M samples (Figure 14). Future work could explore its scalability with larger datasets and tuning durations. Lavender relied on Stable Diffusion v1.4, but adopting higher-resolution models like Stable Diffusion v2 could improve attention map accuracy, albeit with greater resource demands. Additionally, the short inversion steps used to generate attention maps prioritised efficiency but might limit accuracy for unfamiliar words. Lastly, while Lavender was applied to the self-attention-only MiniCPM-v-2.5, further research into optimising self-attention alignment and bridging its differences with cross-attention mechanisms remains an open area for exploration. Full details are available in the Appendix D.

## 9. Conclusion

We have introduced Lavender, a method that leverages the precise text-region alignment of Diffusion Models to enhance Vision-Language Models (VLMs) through efficient supervised fine-tuning. Lavender enables significant performance improvements while remaining highly data-efficient and compute-friendly. Our findings highlight that Lavender effectively aligns VLM attention with Diffusion Models, improving robustness across diverse domains, including challenging and multilingual benchmarks. By incorporating techniques such as parallel attention and LoRA, Lavender balances attention alignment with the preservation of pretrained knowledge, ensuring consistent performance gains without catastrophic forgetting. This scalable, model-agnostic approach demonstrates the promising potential for advancing text-vision alignment in VLMs and lays the groundwork for future research into efficient fine-tuning and cross-modal alignment techniques.

# Impact Statement

**Synergy Between Models: Building a Stronger Vision-Language Ecosystem.** Lavender is a model-agnostic framework that requires no additional human annotations on existing data, making it broadly applicable for enhancing a wide range of VLMs. Given that text–vision alignment is central to most VLM tasks, our attention alignment approach is poised to benefit most downstream applications, as confirmed by extensive benchmark evaluations. Although primarily designed for cross-attention-equipped VLMs, early results show that Lavender also works with self-attention-only models, opening the door to converting widely available LLMs into effective VLMs without retraining. This capability unites the strengths of LLM and diffusion models into a more powerful multimodal expert.

**Data Scarcity.** Both the language and vision communities face looming data shortages, with paired vision–text datasets—the "crude oil" of VLM training—being particularly scarce. End-to-end training from scratch is resource-intensive and often impractical. Large-scale LLMs and DMs have been trained on multi-billion-level datasets, making it inefficient if their knowledge remains isolated. Lavender bridges these models using limited resources—requiring as little as a few thousand samples and one day of training on 8 Nvidia A10G GPUs—enabling small models (below 13B parameters) to perform on par with much larger models (over 50B parameters) across multiple benchmarks.

**Data Privacy.** By leveraging the extensive prior knowledge of diffusion models, Lavender aligns external VLMs with small, local datasets through automatic per-text attention map labelling. This approach is ideal for organisations with limited sensitive data and constrained computational resources, as it enables them to benefit from large-scale pretrained models while keeping data local.

**VLM Attention Alignment with Other Vision Foundation Models.** Lavender enhances text–vision correlation by directly aligning attention within LLM transformer layers. Although our current alignment objectives are derived from Stable Diffusion's attention maps, the same methodology can be applied to other vision foundation models.

**Alternative Multimodal Modalities Beyond Vision–Language.** While our results focus on language and vision, the core idea of aligning cross-attention layers is broadly applicable. Potential applications include text-to-audio alignment (Radford et al., 2023), sequence-to-structure tasks in protein generation (Jumper et al., 2021; Yang et al., 2023), and diverse biomedical applications (Tu et al., 2024; Yang et al., 2024) involving genomics, radiography, pathology, and mammography.

**Attention Alignment as Vision Feedback in Reinforcement Learning.** Our proposed attention alignment loss provides a scalable solution for vision feedback during the RL post-training phase. By leveraging the visual expertise encoded in image generation models, it eliminates the need for costly, fine-grained human annotations and mitigates bias in manual feedback. In effect, it replaces or augments traditional human visual feedback with "diffusion feedback", streamlining the process, reducing training costs, and democratising access to advanced multimodal systems.

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

# Appendix Table of Contents

# A. Extended Key Results

## A.1. Visual Results of Scaled Evaluations on 16 VLM Benchmarks. [Back to Contents]

This section expands Section 6.6. Figure 17 showcases nine VQA examples from multiple evaluated benchmarks, including document, chart, and graphic understanding, real-world OCR, geometry math, visual hallucination, and medical VQA. The comparison highlights the performance of Llama-3.2-11B-Vision-Instruct (Dubey et al., 2024) against its Lavender fine-tuned version. Overall, Lavender demonstrates significantly improved visual understanding compared to the original Llama-3.2 across various VQA tasks, including DocVQA, TextVQA, ChartQA, and WorldMedQA-V. These improvements stem from Lavender's ability to accurately localise and interpret visual elements, leading to better performance on questions requiring detailed visual reasoning. A detailed analysis of these examples is provided in the image caption of Figure 17.

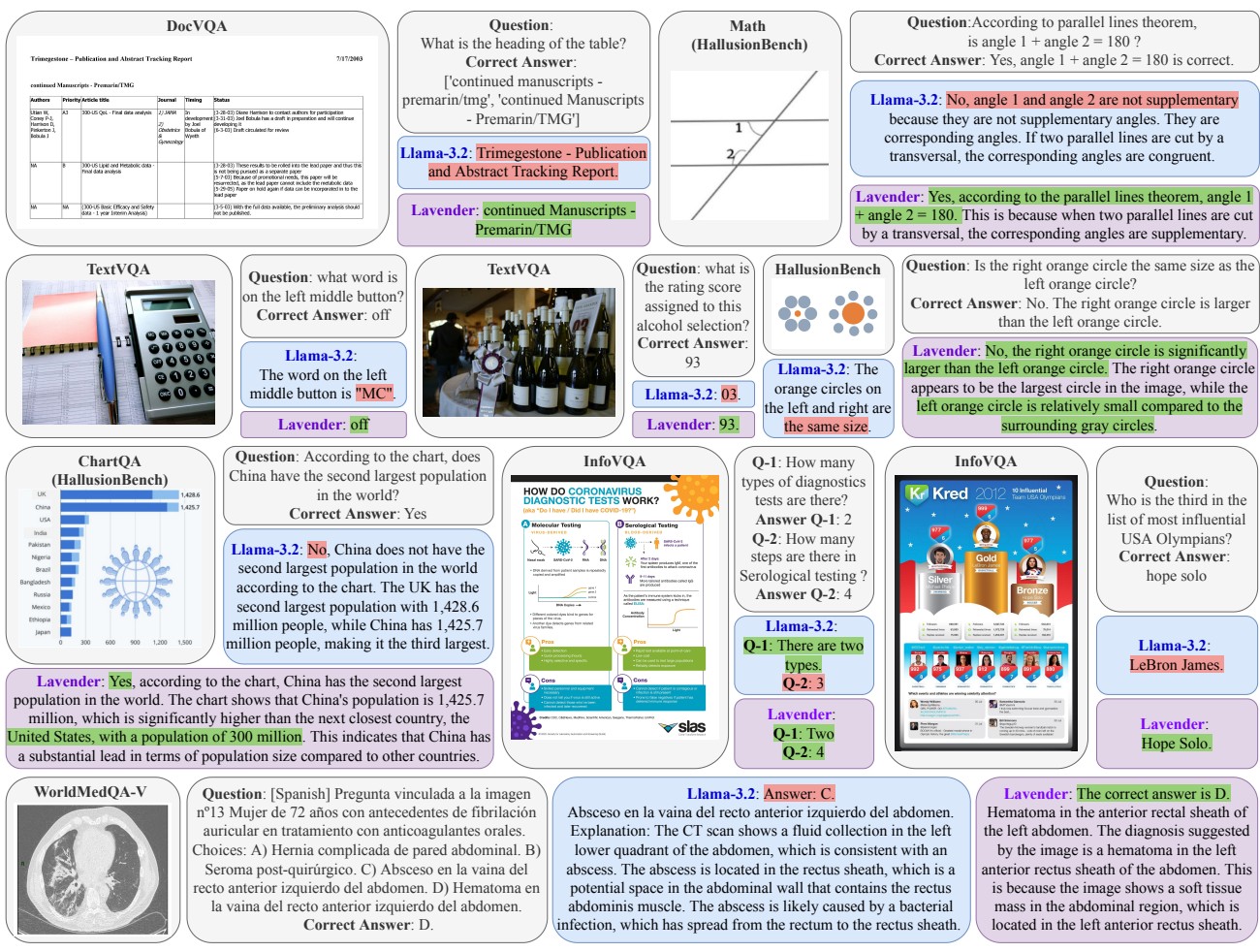

*Figure 17.* **Examples demonstrating Lavender's enhanced fine-granularity vision alignment leading to improved accuracy on various VQA benchmarks.** In the DocVQA example on the top-left, with a question about the title of a table, Llama-3.2 mistakenly extracts the section/page title at the top, which is also in bold, while Lavender correctly identifies the table title located next to the table. Similarly, more accurate visual understanding is directly observed in the two TextVQA examples in the second row. Impressively, the example from HallucinationBench demonstrates Lavender's deep visual understanding of geometry, size, and spatial location, leading to robust anti-hallucination behaviour compared to the original Llama-3.2. In the ChartQA and two InfoVQA examples in the third row, Lavender exhibits its ability to recognise more detailed information, including relatively small elements within the graphs. In the WorldMedQA-V example, which is out-of-distribution, both Llama-3.2 and Lavender understand the question posed in a small language (Spanish) and answer in English. However, Llama-3.2 fails to provide the correct answer due to less accurate visual localization of the unhealthy region. It incorrectly identifies the region as the 'rectus sheath,' which surrounds the actual area of interest, leading to an incorrect response. In contrast, Lavender accurately locates the 'soft tissue mass', resulting in the correct answer.

| Model | Base | FT Data | AI2D | CCBench | DocVQA | InfoVQA | MMBench$^{EN}$ | MME | MMMU-val | MMStar | OCRBench | OCRVQA$^{TQ}$ | POPE | R.W.QA | SEED-IMG | ScienceQA | TextVQA | Hallu.Bench |
|---|---|---|---|---|---|---|---|---|---|---|---|---|---|---|---|---|---|---|
| **Small Budget-Constrained Models (self-attention only)** | | | | | | | | | | | | | | | | | | |
| LLaVA-1.5-7B | Vicuna-7B | 0.15M | 55.5 | 17.8 | 28.1 | 25.8 | 66.5 | 1510.0 | 35.7 | 33.1 | 318.0 | 60.6 | 86.1 | 54.8 | 58.6 | 69.2 | 58.2 | 27.6 |
| LLaVA-NeXt-7B | Vicuna-7B | 0.76M | 67.0 | 24.3 | 74.4 | 37.1 | 67.4 | 1519.0 | 37.6 | 37.6 | 532.0 | 63.8 | 87.5 | 57.8 | 70.2 | 70.3 | 64.9 | 27.6 |
| Mini-Gemini-7B | Qwen-7B | 1.5M | - | - | - | - | 65.8 | 1523.0 | 36.8 | - | 477.0 | - | - | - | - | 71.1 | 65.2 | - |
| Cambrian-1-7B | Vicuna-7B | 10M | 74.6 | 23.7 | 47.9 | 40.8 | 74.6 | 1802.9 | 41.8 | **50.7** | 614.0 | 66.0 | 86.4 | 66.0 | 73.3 | 81.0 | 77.1 | 30.6 |
| Eagle-X5-7B | Vicuna-7B | 0.93M | 73.6 | 28.4 | **86.6** | - | 68.8 | 1866.0 | 37.6 | 41.7 | 551 | 64.3 | **89.3** | 63.8 | 73.6 | 71.2 | 71.9 | 35.4 |
| LLaVA-1.5-8B | Lama3-8B | 0.15M | 69.9 | 27.8 | 32.4 | 27.5 | - | 1825.5 | 39.2 | 46.1 | 420.0 | 61.0 | 87.3 | 56.7 | 70.1 | 72.2 | - | 28.7 |
| LLaVA-Next-8B | Lama3-8B | 0.76M | 72.8 | 32.7 | 78.5 | 38.2 | 74.8 | 1908 | 43.1 | 43.9 | 531.0 | 60.7 | 87.1 | 58.4 | 72.5 | 73.1 | 65.3 | 33.1 |
| Cambrian-1-8B | Lama3-8B | 10M | 73.0 | - | 77.8 | 42.6 | 75.9 | 1547.0 | 42.7 | 50.7 | 624.0 | 66.0 | 73.0 | 64.2 | **74.7** | 73.1 | 71.7 | 30.6 |
| MiniCPM-V-2.5 | Lama3-8B | 0.08M | **78.1** | 45.5 | 84.6 | 52.1 | 76.4 | 2009.1 | 43.1 | 50.3 | 718.0 | 68.7 | 86.6 | 63.9 | 71.9 | 88.8 | 76.6 | 41.9 |
| +Lora FT | Lama3-8B | 0.08M | 77.9 | 44.9 | 84.6 | 52.2 | 76.4 | **2014.2** | 43.1 | 50.6 | 717.0 | 68.8 | 86.2 | 64.1 | 71.9 | 88.9 | 76.6 | 41.8 |
| +Lavender FT | Lama3-8B | 0.08M | 77.9 | **46.3** | 84.6 | **52.3** | 76.9 | 1990.2 | **45.0** | 50.4 | **721.0** | **69.0** | 86.5 | **64.3** | 72.1 | **89.7** | 76.9 | **42.0** |
| **Llama-3.2-11B (cross-attention)** | | | | | | | | | | | | | | | | | | |
| Llama-3.2-11B | LM32-11B | N.A. | 78.7 | 30.6 | 82.8 | 59.0 | 70.8 | 1692.9 | 48.0 | 48.3 | 754.0 | 67.7 | 86.3 | 61.4 | 72.9 | 83.6 | 80.8 | 36.3 |
| + AutoR. Lora-FT | LM32-11B | 0.13M | 77.8 | **39.8** | 86.0 | 61.0 | 73.6 | 1664.6 | 43.7 | 45.1 | 733.0 | 70.0 | 86.9 | 60.7 | 73.6 | 74.6 | 81.1 | 36.6 |
| + Lavender Lora-FT | LM32-11B | 0.13M | **79.0** | 39.2 | **90.3** | **65.1** | **81.3** | **1871.5** | 46.3 | 48.7 | **764.0** | **71.5** | **88.1** | 62.1 | **74.1** | **84.7** | **88.1** | **41.3** |
| + AutoR. Full-FT | LM32-11B | 0.13M | 76.7 | 37.6 | 89.8 | 64.3 | 77.5 | 1697.4 | 45.6 | 46.8 | 748.0 | 68.6 | 87.3 | 59.4 | 71.8 | 77.1 | 81.5 | 37.7 |
| + Lavender Full-FT | LM32-11B | 0.13M | 78.7 | 38.6 | 81.6 | 65.0 | 80.0 | 1695.9 | **49.1** | **49.8** | 686.0 | 70.1 | 87.6 | **62.2** | 73.1 | 84.3 | 84.9 | 39.5 |
| **Small Data-Heavy SOTA Models (<20 B) with Massive FT Data (≥5M)** | | | | | | | | | | | | | | | | | | |
| L.OneVision-7B | Qwen-7B | 5.2M | 82.4 | 54.9 | 87.5* | 68.8* | 83.2 | 1993.6 | 47.9 | **61.9** | 622.0 | 64.7 | **88.4** | 69.9 | 76.7 | 95.4 | 78.3* | 31.6 |
| InternVL2-8B | InternVL2-8B | 5M | 83.6 | **77.1** | 91.6* | 74.8* | 81.7* | 2215.1 | 51.2 | 61.5 | 794.0 | 42.6 | 84.2 | 64.2 | 75.4 | **97.1** | 77.4* | 45.0 |
| Qwen2-VL-7B | Qwen2-7B | ~50M | 83.0 | 65.7 | **94.5*** | **76.5*** | **83.0*** | **2276.3** | 53.7 | 60.7 | **843.0** | 67.5 | **88.4** | 68.5 | 76.0 | 85.5 | **84.3*** | 50.4 |
| Molmo-7B-O | Qwen2-7B | ~35M | 90.7 | 20.6 | 90.8 | 70.0 | 69.1 | 1714.7 | 39.3 | 50.1 | 666.0 | 15.1 | 86.7 | 67.5 | 72.7 | 88.8 | 80.4 | 42.5 |
| Pixtral-12B | Nemo-12B | N.A. | 79.0 | 37.6 | 90.7 | 50.8 | 77.9 | 1921.7 | 52.5 | 54.5 | 685.0 | 64.7 | 84.2 | 65.4 | 71.5 | 87.2 | 75.7 | 47.0 |
| **Large State-of-the-Art Models (>20 B) with Massive FT Data (≥5M)** | | | | | | | | | | | | | | | | | | |
| Cambrian-1-34B | Yi-34B | 10M | 79.7 | 49.2 | 75.5 | 46.0 | 81.4 | 1689.0 | 49.7 | 54.2 | 600.0 | 68.2 | 79.7 | 67.8 | 75.3 | 76.8 | 76.7 | 41.6 |
| L.OneVision-72B | Qwen2-72B | 5.2M | 85.6 | 63.9 | 91.3 | 74.9 | 85.8 | 2257.4 | 56.8 | 65.8 | 741.0 | - | 86.6 | 71.9 | 77.5 | 90.2 | 80.5 | 47.9 |
| Qwen2-VL-72B | Qwen2-72B | ~50M | 88.1 | 69.8 | **96.5*** | 84.5 | 86.5 | **2482.7** | 64.5 | **68.3** | **877.0** | 73.7 | 87.2 | **77.8** | 77.9 | 91.2 | 85.5 | **58.1** |
| Molmo-72B | Qwen2-72B | ~35M | **96.3** | - | 93.5 | 81.9 | 79.4 | 1992.0 | 54.1 | 63.3 | 701.0 | - | 75.2 | - | - | - | 83.1 | 46.6 |
| Claude-3 Haiku | N.A. | N.A. | 86.7 | 24.5 | 88.8 | 56.1 | 60.7 | 1920.0 | 50.2 | 38.1 | 658.0 | - | 74.4 | 45.5 | 63.3 | - | 67.3 | 39.2 |
| Claude-3.5 Sonnet | N.A. | N.A. | 94.7 | 54.1 | 95.2 | 74.3 | 79.7 | 1920.0 | 68.3 | 62.2 | 788.0 | - | 73.6 | 60.1 | 72.2 | 88.9 | 74.1 | 49.9 |
| GPT-4V (0409) | N.A. | N.A. | 89.4 | 57.3 | 87.2 | 75.1 | 81.0 | 2070.2 | 63.1 | 56.0 | 656.0 | - | 81.8 | 61.4 | 73.0 | 84.8 | 78.0 | 43.9 |
| GPT-4o (0513) | N.A. | N.A. | 94.2 | **71.2** | 92.8 | 79.2 | 83.4 | 2310.3 | **69.1** | 63.9 | 736.0 | - | 85.6 | 75.4 | 77.1 | 90.7 | 77.4 | 55.0 |
| Gemini 1.5 Pro | N.A. | N.A. | 94.4 | 28.4 | 93.1 | 81.0 | 73.9 | 2110.6 | 62.2 | 59.1 | 754.0 | 12.3 | **88.2** | 64.1 | 76.0 | 85.7 | 78.7 | 55.9 |
| Llama-3.2-90B | Llama-3.1-70B | N.A. | 92.3* | 54.1 | 85.7 | - | 80.4 | 1741.0 | 60.3 | 55.3 | 783.0 | - | 86.3 | 68.2 | 76.8 | 87.1 | - | 44.1 |

*Table 1.* **Zero-shot accuracy of various fine-tuned models across 16 VLM benchmarks.** Results are grouped into four sections based on base model size and the scale of the fine-tuning dataset. The top score for each benchmark within each group is highlighted in bold. Scores for MiniCPM-V-2.5, Llama-3.2-11B, and their autoregressive (AutoR.) and Lavender fine-tuned variants are locally evaluated using OpenCompass Vlmevalkit (Duan et al., 2024) with 'gpt-4o' as the evaluator. All other scores are sourced from the OpenCompass Multi-Modal Leaderboard, evaluated with the same Vlmevalkit. When leaderboard results are unavailable, the models' published numbers are cited, marked with a * notation if provided. Our observations are categorized as follows: 1) *Small Budget-Constrained Models.* This group is most comparable to Lavender in terms of parameter size and fine-tuning data scale. Both Lavender versions outperform the majority of benchmarks with significant margins over the second-best-performing external models, achieving improvements of up to 40% on CCBench. On a few benchmarks (e.g., MMStar, POPE, and SEED-IMG), Lavender is surpassed by baseline models, though the difference is within 4% for SEED-IMG. Lavender-Llama-3.2 occasionally underperforms on MME due to privacy protection constraints (see Table 3). 2) *Small Data-Heavy SOTA Models.* The primary argument for Lavender in this work is its ability to achieve improvements over the autoregressive fine-tuning baseline. Lavender implementations on MiniCPMv2.5 and Llama-3.2-11B are not designed to beat state-of-the-art results due to the limited fine-tuning data scale (0.13M), which is significantly smaller than the 5M to 50M datasets used for this group, approximately 38x to 384x larger than that used for Lavender. We simplify the comparison by excluding pretraining dataset sizes for base models. In this group, Qwen2-VL-7B and LLaVA-OneVision-7B are the top performers, surpassing Lavender on most benchmarks. However, we note that the performance gap is likely influenced by the composition of fine-tuning datasets, as discussed in Section 6.3. 3) *Large SOTA Models.* Finally, we include results from the latest state-of-the-art models, which are at least 20B in size (or unreleased models typically exceeding 100B) and fine-tuned on datasets of at least 5M samples. Despite Qwen2-VL-72B outperforming all other models on 18/20 benchmarks, Lavender pushes the boundaries of Llama-3.2-11B, achieving performance comparable to certain closed-source models that are at least an order of magnitude larger (e.g., Claude-3.5 Sonnet, GPT-4o, and Gemini 1.5 Pro) on benchmarks such as TextVQA, POPE, RealWorld, and DocVQA.

**A.2. Scaling Behaviour.** [Back to Contents]

This section expands on Appendix A.2: *Scaling Behaviour* in the main paper. Lavender is a general model-agnostic approach but evaluated only on small fine-tuning datasets in this work, due to limited computational resources, it is important to assess how it may scale with larger datasets. In this section, we evaluate multiple checkpoints during the fine-tuning of Llama-3.2-11B on four combinations of RV83k, Flk30k, and OV30k datasets, using either autoregressive or Lavender methods, combined with LoRA or full fine-tuning strategies. The results in Figure 18 are based on the average performance across eight benchmarks, with detailed results for each benchmark provided in Figure 28. The findings show that Lavender scales better as more data is sampled and effectively reduces overfitting—a challenge often faced by autoregressive fine-tuning, particularly on small fine-tuning datasets. Additionally, we observe that larger datasets reduce performance variation, an expected behaviour during scaling.

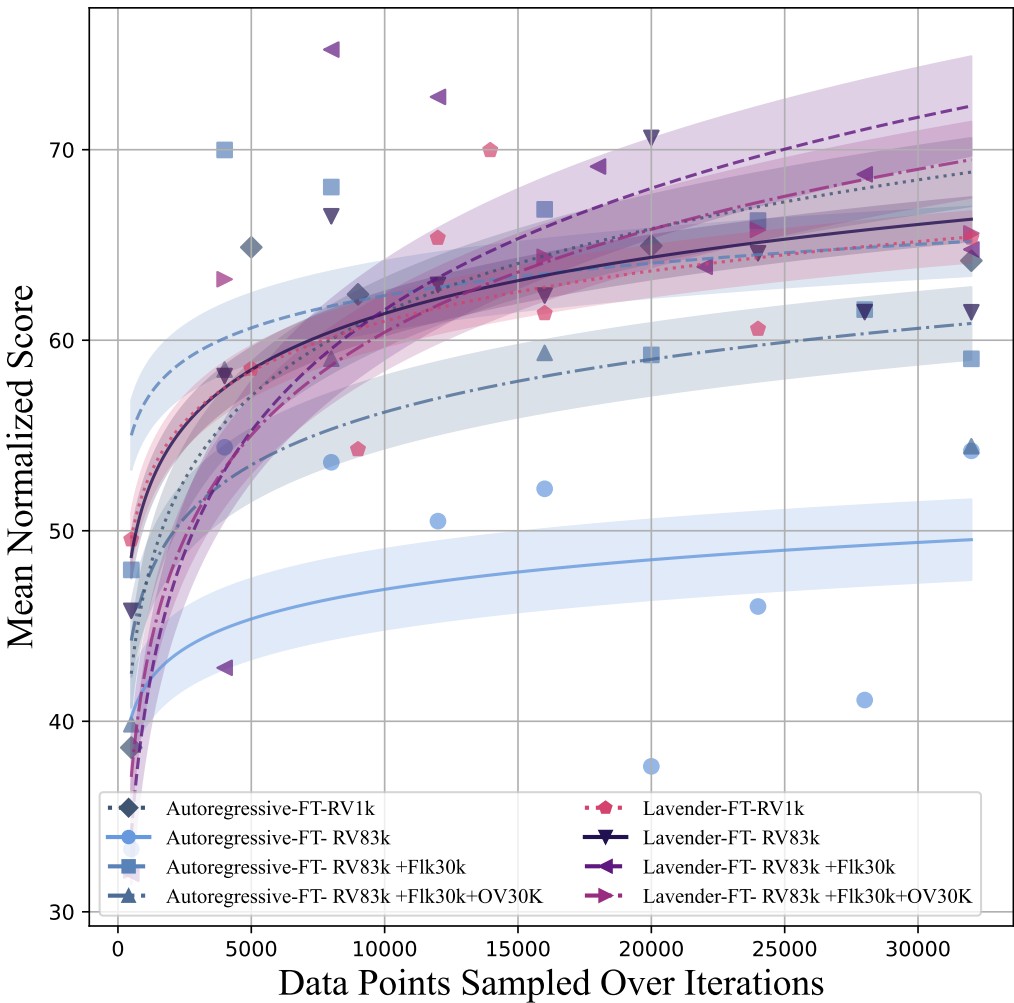

*Figure 18.* **Lavender scales better and reduces overfitting compared to autoregressive fine-tuning, with larger datasets lowering variations.** The plot shows the mean normalised performance of four dataset configurations across eight benchmarks after LoRA and Full fine-tuning of Llama 3.2-11B, as a function of the number of data points sampled. Markers represent observed performance for each method-iteration pair, while trendlines with different styles indicate the overall performance trends. The shaded regions around the trendlines represent confidence intervals derived from the standard error, showing the uncertainty of the trendline predictions. Narrower regions indicate higher confidence, while wider regions suggest greater variability. Per-benchmark results are in Figure 28. The simplified version is presented in Figure 14.

**A.3. Visual Results of Aligned Attention Maps with Llama 3.2-11B.** [Back to Contents]

This section expands on Section 6.6 in the main paper. A key argument of Lavender is its ability to align the attention maps of VLMs to those of DMs. Figure 19 provides examples of the cross-attention maps from Llama 3.2-11B after full fine-tuning with Lavender. The results indicate that the aligned VLM attention maps generally correlate with the semantic regions of corresponding words in a manner similar to the Diffusion Model. Interestingly, the VLM attention maps after alignment are more concentrated than those of the Diffusion Model. This difference is likely driven by their distinct optimization goals: sense understanding for optimal text generation in VLMs versus pixel-level precision for image generation in DMs.

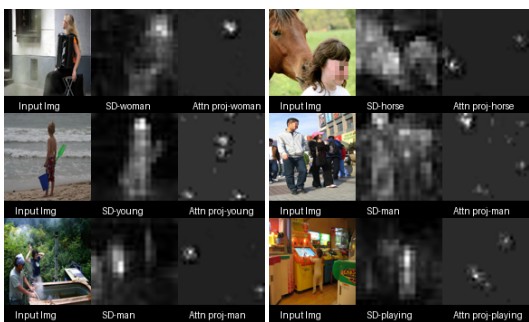

*Figure 19.* **The per-word VLM attention maps are aligned to the Stable Diffusion (SD) after tuning with Lavender.** Results are from Lavender-Llama 3.2-11B implementation. More results are available in Appendix Figure 30.

## B. Ablation and Analysis

This section expands on Section 7: *Ablation and Analysis Summary* in the main paper. We conduct ablation experiments to analyse the key components and strategies of the Lavender framework and discuss their contributions to its performance.

**B.1. Attention Aggregation Functions.** [Back to Contents]

We begin by examining different attention aggregation functions in VLMs, a key component discussed in Section 3. These include simple averaging and maximisation operations over layers, referred to as 'layer-mean' and 'layer-max,' as outlined in Section 3.2.1; attention flow combined with multiplicative and additive aggregations, referred to as 'flow-multi' and 'flow-sum,' as defined in Section 3.2.2; and learned attention aggregation, referred to as 'learn,' as discussed in Section 3.2.3.

**Comparing the Aggregation Functions on Eight Benchmarks.** We fully fine-tune Lavender-Llama 3.2-11B on a small Flickr-1k subset for 20 epochs and evaluate all fine-tuned models on eight benchmarks without using GPT-API as the evaluator but 'exact-match' in the VQA assessments to reduce costs. Note that this approach differs from the main results in Table 1 and leads to an expected performance drop of approximately 10% to 20%.

Results in Figure 20 show that the attention 'flow-multi' aggregation performs the weakest overall, likely due to overly compressing and overwriting information through multiplicative aggregation across all layers. This strategy aims to aggregate the most important region for each image-question pair, where the question includes all words, but it proves less effective for per-word attention alignment. In contrast, the attention 'flow-sum' strategy performs well, ranking second overall. This indicates that the additive operation is better at preserving information during aggregation. Simple averaging and max operations ('layer-mean' and 'layer-max') demonstrate surprising robustness, achieving mid-level performance overall and excelling on specific benchmarks such as MME and InfoVQA. This suggests that these methods avoid biases introduced by explicit ad-hoc aggregation functions like attention 'flow-multi'. Among all results, learned aggregation consistently outperforms other methods, confirming the design principles of *Preserving Pretrained Attention Mechanisms* and *Capturing Complex Semantic Correlations*.

**Scalability of Aggregation Functions.** We further assess the scalability of the proposed aggregation functions by comparing the eight benchmarks' performance at different training lengths. The average results are shown in Figure 21, with detailed results for each individual benchmark provided in Appendix Figure 29. Despite the consistent overall ranking discussed earlier, the 'learn' aggregation strategy demonstrates the best scalability. Starting as the second worst at epoch 5, it climbs to the best-performing method by epoch 20. This behaviour can be understood in light of the limitations of manually

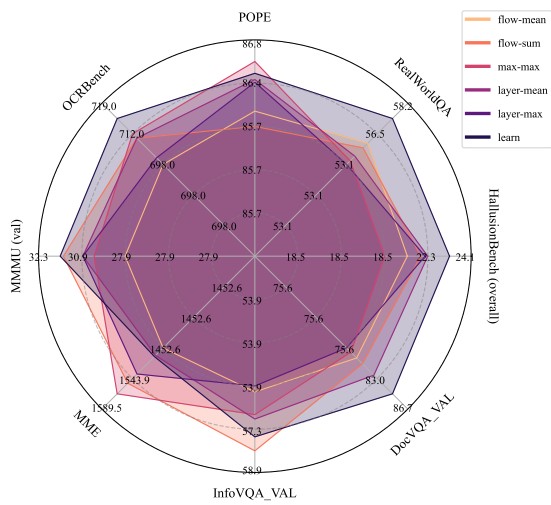

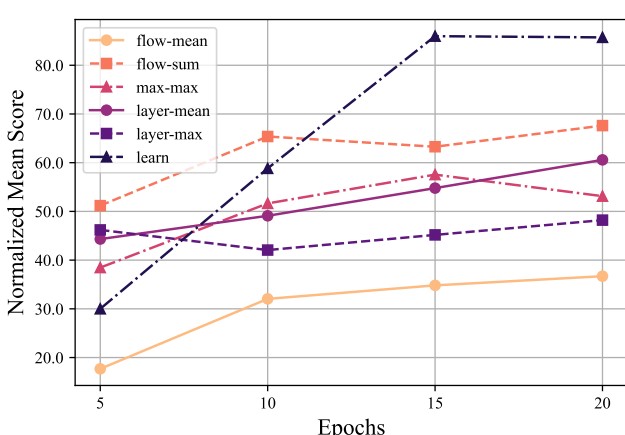

*Figure 20.* **Comparing aggregation functions on eight benchmarks.** Results are from Lavender-Llama 3.2-11B fine-tuned on Flickr-1k for 20 epochs, evaluated with 'exact match' without an LLM judge.

*Figure 21.* **Mean performance over training epochs.** Results from Lavender-Llama 3.2-11B fine-tuned on Flickr-1k. Detailed per-benchmark results are in Appendix Figure 29.

designed aggregations, such as 'flow-sum.' While manual aggregations may initially reflect straightforward text-to-region correlations, like identifying colours in images, they are inherently suboptimal. These methods act as a shortcut, achieving good early performance but lacking scalability when encountering more complex text-to-region correlations. Learnable aggregation, in contrast, adapts to these complexities and scales effectively with additional training.

### B.2. Training recipes. [Back to Contents]

**Pretraining the Aligner Network.**    This work aims to enhance the alignment of pretrained VLMs, such as Llama-3.2, which are already knowledgeable. As discussed in Section 5, fully fine-tuning such VLMs with a secondary attention alignment objective and a small dataset can lead to catastrophic forgetting. The results in the first row of Figure 22 confirm this concern, showing drastic drops in performance across eight evaluated benchmarks after fully fine-tuning on a small dataset for 30 epochs. In contrast, the proposed pretraining strategy significantly mitigates this issue. The pretraining strategy involves pretraining only the Aligner Network for a certain number of epochs while keeping the VLM parameters frozen, before jointly updating all parameters. Results in Figure 22 show that pretraining for a short duration (less than one-third of the total epochs) is particularly effective overall. Notably, more challenging benchmarks that require deeper interaction between visual perception, complex reasoning, and domain knowledge, such as MMMU and RealWorldQA, benefit more from longer pretraining.

**Full Finetuning Versus LoRA.**    Comparing full finetuning with the LoRA finetuning strategy for Lavender, our main results with Llama-3.2 in Figure 11 and Table 1 suggest that LoRA is generally more beneficial for the majority of benchmarks. However, we also observe that full finetuning outperforms LoRA with Lavender on a few more challenging benchmarks, such as MMMU, MMStar, and RealWorldQA, which require deeper interactions between visual perception, complex reasoning, and domain knowledge. This observation is consistent with the results from the ablation study in Figure 22. Given that this work focuses on short finetuning overall, these findings suggest that *LoRA finetuning with Lavender offers better short-term benefits*. In contrast, *scaling Lavender with full finetuning may lead to deeper alignment and knowledge restructuring, potentially resulting in longer-term advantages*.

**Choice of Layers to Align.**    We examine the choice of cross-attention layers to align with Lavender in the Llama 3.2-11B Vision Instruct model. This model comprises 40 layers in total, 8 of which are cross-attention layers integrated to process visual inputs, constituting one-fifth of the model's layers. Figure 23 compares the performance of four different subsets of the 8 cross-attention layers. While attending to the first, mid, or last subset shows shifted strengths on specific benchmarks—RealWorldQA/OCRBench, MMMU/POPE, and Hallucination, respectively—aligning all 8 layers proves to be the most effective overall. Therefore, this is the default strategy adopted in this work.

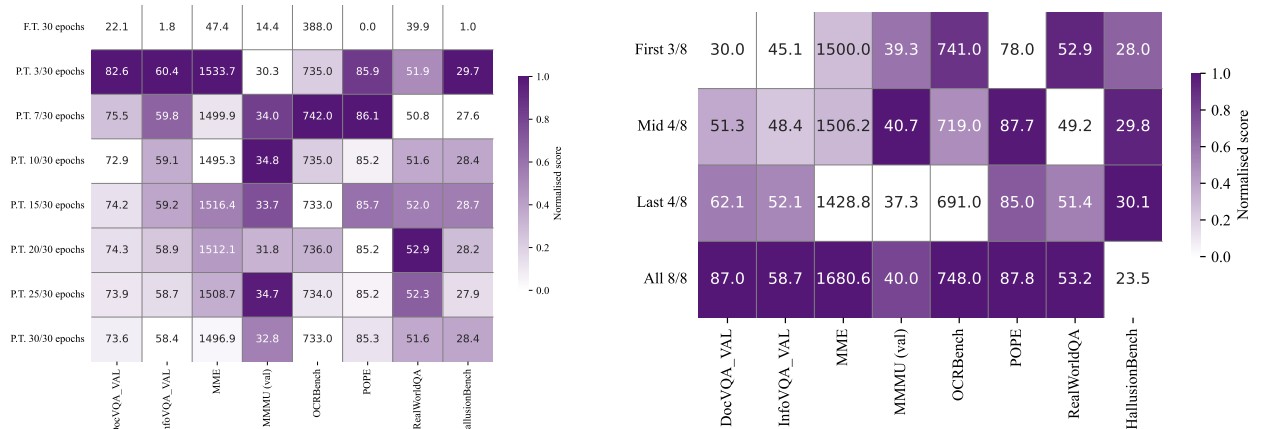

*Figure 22.* **Short pretraining (P.T.) of the Aligner Network mitigates catastrophic forgetting.** Results are from Lavender-Llama 3.2-11B fine-tuned on Flickr-1k for 30 epochs, with varying pretraining lengths, evaluated on eight benchmarks using 'exact match' without an LLM judge.

*Figure 23.* **Aligning all eight cross-attention layers in Llama-3.2 is most effective.** Results compare subsets of cross-attention layers aligned with Lavender in Llama 3.2-11B, LoRA fine-tuned on Flickr-1k for 30 epochs, evaluated on eight benchmarks using 'exact match' without an LLM judge.

## C. Failure Strategies [Back to Contents]

This section expands on Section 8: *Failure Strategies* in the main paper. Despite the successful strategies presented earlier, we also tested the following approaches and found them to be less effective: 1) *Fully Finetuning Without Pretraining.* As shown in Figure 22, fully finetuning without pretraining often destabilises the model, while LoRA proves to be generally robust. We recommend starting with LoRA on small datasets as a reliable initial strategy for exploring Lavender, reserving full finetuning for scaling efforts and investigating emerging behaviours after deep alignment. 2) *Frequent Switching Between Training Strategies.* Given the varied performance of certain Lavender strategies, such as long fully finetuning or aligning subsets of cross-attention layers, we experimented with more complex staged training strategies. These involved switching between different strategies across epochs. However, frequent strategy changes harmed performance, likely due to the short overall training length. The model appeared to struggle with prioritizing objectives across strategies before stabilizing, leading to crashes. 3) *Mixing Additional Data.* In some cases, adding extra data did not improve performance. For example, as shown in Figure 18, mixing the OCRVQA dataset reduced overall performance. This suggests that OCRVQA may lead to overfitting more readily than other datasets tested.

## D. Limitation and Future Works [Back to Contents]

This section expands on Section 8: *Limitation and Future Works* in the main paper.

**Limited Compute.** Lavender was evaluated on datasets of up to 0.13M samples, constrained by available compute resources. This is significantly smaller than the 5M to 50M datasets used by state-of-the-art models, which are approximately 38x to 384x larger. Figure 18 demonstrates non-convergent scaling behaviour, suggesting that further scaling of both dataset size and tuning length could lead to additional improvements in overall performance with Lavender.

**Exploring Higher-Resolution Diffusion Models.** Lavender is evaluated using Stable Diffusion v1.4 (Rombach et al., 2021) in this work. Advanced models, such as Stable Diffusion v2 (Rombach et al., 2022), could provide higher resolution and more accurate attention maps but would demand greater memory for attention alignment, exceeding the capacity of this study. This approach may become viable with sufficient data scaling and tuning as task complexity increases, marking a key direction for future research.

**More Accurate Attention Maps Extractions.** This work focuses on verifying the feasibility of Lavender rather than the quality of attention maps from Diffusion Models, which already exhibit significantly stronger text-region alignment. However, as noted earlier, short inversion and diffusion steps were applied to prepare per-word attention maps, reducing

processing time to 20 seconds per image on a single V100 GPU. While this setup is efficient, it may limit the accuracy of text-to-region correlations in the attention maps, especially for unfamiliar words (Jin et al., 2023). Future work could explore efficient methods to accelerate attention map estimation, enhancing Lavender's overall effectiveness.

**Improved Handling of Self-Attention Only Models.**   This work demonstrated the feasibility of applying Lavender to the self-attention-only Llama-3-based MiniCPM-v-2.5. However, further optimization for such models remains under-explored due to limited capabilities and computational resources. Notably, this includes the advanced parallel attention mechanism, which was proposed and shown to be effective in the Llama-3.2 implementation of Lavender. Future theoretical and empirical research on the relationship and translatability between self-attention and cross-attention mechanisms would be valuable for advancing this line of inquiry and benefiting the broader community.

## E. Related Work [Back to Contents]

This section expands on Section 1: *Introduction* in the main paper.

*We find that the gap in VLM alignment partly stems from the technological trajectories pursued over the past half-decade.* One key VLM milestone was Flamingo (Alayrac et al., 2022), which laid the foundation for modern VLMs. In its design, images and text are processed by separate encoders, unified through a perceiver resampler (Jaegle et al., 2021), and passed through deep transformer layers combining cross-attention and self-attention. Flamingo's elegant architecture established a new standard, influencing a range of subsequent models (Li et al., 2022; 2023b; You et al., 2023). The importance of aligning vision-text correlations is evident in the design of Llama 3.2 (Dubey et al., 2024), released two years later, which adopts Flamingo's strategy of using a dedicated cross-attention module for effective interaction handling. However, training a VLM with a dedicated cross-attention module end-to-end requires substantial data and computational resources. These models are typically pre-trained on millions or even billions of image-text pairs and interleaved image-text datasets (Zhu et al., 2024). Similar challenges apply to broader multimodal models beyond vision and language (Lu et al., 2024).

Unlike VLMs, single-modality large language models (LLMs) have scaled more rapidly (Ouyang et al., 2022; Brown et al., 2020; Chowdhery et al., 2023), often consuming over 100 million examples spanning 1,800 tasks (Longpre et al., 2023). VLMs, however, face a training data gap due to the high cost of acquiring paired image-text datasets. To address this, researchers proposed leveraging scaled LLMs by instruction fine-tuning them on as little as 150k paired visual question answering (VQA) data using an autoregressive loss. This approach, pioneered by Zhu et al. (2023); Dai et al. (2023); Liu et al. (2024c), aligns text and image tokens through fine-tuning connectors such as MLPs, encoders, or decoders connecting to the LLM, providing an efficient pathway to integrate vision with language models for diverse tasks (Wang et al., 2024a; Li et al., 2024; Koh et al., 2024; Chen et al., 2025a; Wang et al., 2024b; Chen et al., 2023; Huang et al., 2024; Liu et al., 2024d; Gao et al., 2023; Cha et al., 2024).

However, the community soon recognised that the vision capabilities integrated through these small adapter layers outside the LLM (in LLaVA-like approaches) remain insufficient (Tong et al., 2024c). To address this gap, Covert et al. (2024); Karamcheti et al. (2024) refined vision encoders to align more closely with pretrained vision models. Jiang et al. (2023); Kar et al. (2025) proposed merging multiple visual encoders with projection layers before feeding them into the LLM. Separately, Tong et al. (2024a); Shi et al. (2024); Zong et al. (2024) explored merging a larger number of diverse vision expert models with fine-tuned projection layers, integrating the combined vision features either before the LLM or within the LLM transformer layers, respectively.

Diffusion Models (DMs), as a key vision expert, have also garnered recent attention. Wang et al. (2024c) leverage DM's image generation loss to enhance the visual encoder. Other approaches focus on equipping VLMs with DM's image generation capabilities, either by using VLM outputs to fine-tune DMs separately (Tong et al., 2024b; Hernandez et al., 2024), sharing a central transformer (Shi et al., 2025; Chen et al., 2025b) or by directly merging DMs with LLM transformers (Zhou et al., 2024).

Despite attempts to integrate DMs and VLMs with minimal modifications to their internal architectures, *one overlooked aspect in prior work is the role of self- and cross-attention layers within DM and LLM Transformers*. These layers govern the interplay between multi-modal tokens, yet have not been closely examined. Although DMs and the LLM component in VLMs share the same foundational Transformer architecture (Vaswani, 2017; Dosovitskiy, 2020), they exhibit markedly different text-to-region alignment due to their distinct optimization objectives. Notably, DMs demonstrate stronger alignment than VLMs, as shown in Figure 3.

# F. Full Pseudo Code of Diffusion Instruction Tuning [Back to Contents]

This section expands on Algorithm 1: *Algorithm 1*, as introduced in the main paper.

---

**Algorithm 2** Diffusion Instruction Tuning (Full Version)

---

**Require:** Dataset $D = \{(x^{(i)}, y^{(i)}, y_q^{(i)}, y_l^{(i)})\}_{i=1}^N$, where $y_q^{(i)}$ is textual input (e.g. a question) and $y_l^{(i)}$ is the target textual output. Pretrained DM parameters $\theta_D$, pretrained VLM parameters $\theta$. Scaling factor $\lambda > 0$, learning rate $\eta$.

**Ensure:** Fine-tuned VLM parameters $\theta$

  1: **Stage 1: Preprocessing (Run Once)**
     *# the DM processes image-question pairs, hence replacing $y_q$ with $y$*
  2: For each data point $(x^{(i)}, y^{(i)})$ in $D$:
  3:    Use the pretrained DM ($\theta_D$ fixed) to compute $p_{\text{DM}}(a \mid x^{(i)}, y^{(i)}; \theta_D)$
  4:    Store these attention distributions: $A_{\text{DM}}^{(i)} \leftarrow p_{\text{DM}}(a \mid x^{(i)}, y^{(i)}; \theta_D)$
  5: After this, we have a DM-derived attention target for each data point, which will remain fixed during fine-tuning.

  6: **Stage 2: Fine-Tuning the VLM**
  7: Initialise $\theta$ from a pretrained VLM.
  8: Set a learning rate $\eta$ and define maximum training steps or convergence criteria.
  9: **repeat**
10:    Sample a mini-batch $B \subseteq D$ of size $m$
11:    $L_{\text{VLM}}(\theta) \leftarrow 0, L_{\text{att}}(\theta) \leftarrow 0$
12:    **for** $(x^{(j)}, y^{(j)}, y_q^{(j)}, y_l^{(j)})$ in $B$ **do**
13:      Compute $p_{\text{VLM}}(a \mid x^{(j)}, y^{(j)}; \theta)$ by aggregating attention heads/layers of the VLM cross-/self-attention (see Section 3)
14:      $\delta^{(j)}(\theta) \leftarrow \text{Aligner}\big(p_{\text{VLM}}(a \mid x^{(j)}, y^{(j)}; \theta)\big) - A_{\text{DM}}^{(j)}$
15:      Compute task loss: $L_{\text{VLM}}(\theta) \mathrel{+}= -\log p(y_l^{(j)} \mid x^{(j)}, y_q^{(j)}; \theta)$
16:      Compute attention alignment loss: $L_{\text{att}}(\theta) \mathrel{+}= \|\delta^{(j)}(\theta)\|^2$
17:    **end for**
18:    Form total loss: $L_{\text{total}}(\theta) = L_{\text{VLM}}(\theta) + \lambda L_{\text{att}}(\theta)$
19:    Update parameters: $\theta \leftarrow \theta - \eta \nabla_\theta L_{\text{total}}(\theta)$
20:    Optionally, apply LoRA or other optimization tricks if desired.
21: **until** convergence (e.g., validation metric plateaus or max steps reached)
22: **Output:** The fine-tuned VLM parameters $\theta$.

---

# G. Bayesian Justification for DM Attention Proximity [Back to Contents]

This section expands on Section 2.3: *Bayesian Derivation*, as discussed in the main paper. In this section, we provide a detailed Bayesian justification for our assumption that the DM's attention distribution $p_{\text{DM}}(a \mid x, y; \theta_D)$ is closer to the optimal posterior distribution of vision-centric word-to-region attention $p^*(a \mid x, y)$ than the VLM's attention distribution $p_{\text{VLM}}(a \mid x, y; \theta)$.

## G.1. Modelling Attention Distributions as Posteriors

We model the attention distributions in both the DM and the VLM as posterior probabilities over the attention $a$ given the inputs and model parameters. For consistency, we consider the joint distributions and apply Bayes' theorem.

**Diffusion Model (DM):** The DM is trained to generate an image $x$ conditioned on text $y$ by modelling the distribution $p_{\text{DM}}(x \mid y; \theta_D)$. We can express the attention mechanism in the DM as contributing to this distribution via:

$$p_{\text{DM}}(x \mid y; \theta_D) = \int p_{\text{DM}}(x \mid y, a; \theta_D) \, p_{\text{DM}}(a \mid y; \theta_D) \, da. \tag{17}$$

Applying Bayes' theorem, the posterior over attention $a$ given $x$ and $y$ is:

$$p_{\text{DM}}(a \mid x, y; \theta_D) = \frac{p_{\text{DM}}(x \mid y, a; \theta_D)\, p_{\text{DM}}(a \mid y; \theta_D)}{p_{\text{DM}}(x \mid y; \theta_D)}. \tag{18}$$

**Vision-Language Model (VLM):** Similarly, for the VLM, which generates textual output $y_l$ given an image $x$ and textual input $y_q$, the attention mechanism influences the distribution $p_{\text{VLM}}(y_l \mid x, y_q; \theta)$:

$$p_{\text{VLM}}(y_l \mid x, y_q; \theta) = \int p_{\text{VLM}}(y_l \mid x, y_q, a; \theta)\, p_{\text{VLM}}(a \mid x, y_q; \theta)\, da. \tag{19}$$

The posterior over attention $a$ is then:

$$p_{\text{VLM}}(a \mid x, y_q, y_l; \theta) = \frac{p_{\text{VLM}}(y_l \mid x, y_q, a; \theta)\, p_{\text{VLM}}(a \mid x, y_q; \theta)}{p_{\text{VLM}}(y_l \mid x, y_q; \theta)}. \tag{20}$$

### G.2. Differences in Likelihood Functions

The key distinction arises from the likelihood functions:

- **DM Likelihood $p_{\text{DM}}(x \mid y, a; \theta_D)$:** The DM must reconstruct the image $x$ accurately, which requires precise alignment between textual tokens and visual features. The attention $a$ plays a critical role in ensuring that each part of the text $y$ correctly influences the corresponding visual content in $x$.

- **VLM Likelihood $p_{\text{VLM}}(y_l \mid x, y_q, a; \theta)$:** The VLM generates text $y_l$ based on the image $x$ and input text $y_q$. While attention $a$ aids in focusing on relevant visual regions, the text generation process can often rely on higher-level visual features and may not require as fine-grained vision-text alignment as the DM.

### G.3. Entropy and Concentration of Attention Distributions

Due to the DM's need for precise image reconstruction, its attention distribution $p_{\text{DM}}(a \mid x, y; \theta_D)$ is expected to be more concentrated around $p^*(a \mid x, y)$. This can be quantified by the entropy $H$ of the attention distributions:

$$H\left(p_{\text{DM}}(a \mid x, y; \theta_D)\right) < H\left(p_{\text{VLM}}(a \mid x, y_q; \theta)\right). \tag{21}$$

A lower entropy indicates that the DM's attention distribution is more peaked and thus closer to $p^*(a \mid x, y)$, whereas the VLM's higher entropy reflects a more diffuse attention distribution. **This is consistent with our empirical observations in Figure 3 and section 6.1**.

### G.4. KL Divergence to the Ideal Attention in Vision-Centric Tasks

We can formalise the proximity to the optimal posterior attention distribution using the Kullback-Leibler (KL) divergence. For the DM conditioned on a unified text $y$, modeling $p(x|y; \theta_D)$, the KL divergence to the $p^*(a \mid x, y)$ is:

$$D_{\text{KL}}\left(p_{\text{DM}}(a \mid x, y; \theta_D) \,\|\, p^*(a \mid x, y)\right) = \int p_{\text{DM}}(a \mid x, y; \theta_D) \log \frac{p_{\text{DM}}(a \mid x, y; \theta_D)}{p^*(a \mid x, y)}\, da, \tag{22}$$

and similarly, for the VLM processes an image $x$, question $y_q$, and answer label $y_l$, modeling $p(y_l|x, y_q; \theta)$:

$$D_{\text{KL}}\left(p_{\text{VLM}}(a \mid x, y_q; \theta) \,\|\, p^*(a \mid x, y)\right) = \int p_{\text{VLM}}(a \mid x, y_q; \theta) \log \frac{p_{\text{VLM}}(a \mid x, y_q; \theta)}{p^*(a \mid x, y)}\, da. \tag{23}$$

Bringing in the entropy $H$ of the attention distributions, equations (22) and (23) can be written as:

$$D_{\text{KL}}\left(p_{\text{DM}}(a \mid x, y; \theta_D) \parallel p^*(a \mid x, y)\right) = -H\left(p_{\text{DM}}(a \mid x, y; \theta_D)\right) - \int p_{\text{DM}}(a \mid x, y; \theta_D) \log p^*(a \mid x, y)\, da, \quad (24)$$

and similarly for the VLM:

$$D_{\text{KL}}\left(p_{\text{VLM}}(a \mid x, y; \theta) \parallel p^*(a \mid x, y)\right) = -H\left(p_{\text{VLM}}(a \mid x, y; \theta)\right) - \int p_{\text{VLM}}(a \mid x, y; \theta) \log p^*(a \mid x, y)\, da. \quad (25)$$

Our empirical observation of lower entropy in equation (21) indicates that the DM's attention distribution is more concentrated and thus more certain about where to attend, which is a consequence of the DM's need for precise vision-text alignment during image reconstruction.

Assuming that the cross-entropy terms $\int p_{\text{DM}}(a \mid x, y; \theta_D) \log p^*(a \mid x, y)\, da$ and $\int p_{\text{VLM}}(a \mid x, y; \theta) \log p^*(a \mid x, y)\, da$ are *approximately equal*[1] or that the difference in entropies dominates the difference in cross-entropies, we can infer:

$$D_{\text{KL}}\left(p_{\text{DM}}(a \mid x, y; \theta_D) \parallel p^*(a \mid x, y)\right) < D_{\text{KL}}\left(p_{\text{VLM}}(a \mid x, y; \theta) \parallel p^*(a \mid x, y)\right). \quad (26)$$

This inequality suggests that the DM's attention distribution is closer to the optimal posterior attention distribution in terms of KL divergence. The lower entropy of the DM's attention implies it is more peaked around the ideal attention in vision-centric tasks. Therefore, our empirical observation of the DM's lower attention entropy supports the assumption that the DM's attention distribution is closer to the optimal posterior attention distribution than the VLM's. By aligning the VLM's attention with the DM's attention, we aim to reduce the VLM's KL divergence to the ideal attention in vision-centric tasks, enhancing its vision-text alignment and overall performance.

## H. Justification of the Attention Alignment Loss [Back to Contents]

This section expands on Equation (10), as introduced in the main paper, to provide a detailed justification for the inclusion of the attention alignment loss $L_{\text{att}}(\theta)$ and the scaling factor $\lambda$ in our Bayesian framework.

### H.1. Derivation of the Likelihood Term

We start by modelling the difference between the VLM's attention distribution and the DM's attention distribution as a random variable. For each data point $i$, we define:

$$\delta^{(i)}(\theta) = p_{\text{VLM}}(a \mid x^{(i)}, y^{(i)}; \theta) - p_{\text{DM}}(a \mid x^{(i)}, y^{(i)}; \theta_D). \quad (27)$$

We assume that $\delta^{(i)}(\theta)$ follows a multivariate normal distribution with zero mean and covariance matrix $\sigma^2 I$, where $I$ is the identity matrix:

$$\delta^{(i)}(\theta) \sim \mathcal{N}(0, \sigma^2 I). \quad (28)$$

Under this assumption, the probability density function for $\delta^{(i)}(\theta)$ is:

$$p\left(\delta^{(i)}(\theta)\right) = \frac{1}{(2\pi\sigma^2)^{k/2}} \exp\left(-\frac{1}{2\sigma^2} \left\|\delta^{(i)}(\theta)\right\|^2\right), \quad (29)$$

where $k$ is the dimensionality of the attention distribution.

The likelihood of observing the DM's attention given the VLM's parameters over the entire dataset is then:

$$p(A_{\text{DM}} \mid \theta) = \prod_i p\left(\delta^{(i)}(\theta)\right) = \left(\frac{1}{(2\pi\sigma^2)^{k/2}}\right)^N \exp\left(-\frac{1}{2\sigma^2} \sum_i \left\|\delta^{(i)}(\theta)\right\|^2\right), \quad (30)$$

---

[1]Since $p^*(a \mid x, y)$ is the same for both models, and both $p_{\text{DM}}$ and $p_{\text{VLM}}$ are centered around $p^*(a \mid x, y)$ given they are from pretrained models based on big dataset, the values of these cross-entropy terms would not differ significantly.

where $N$ is the number of data points.

Taking the negative log-likelihood, we get:

$$-\log p(A_{\mathrm{DM}} \mid \theta) = \frac{Nk}{2}\log(2\pi\sigma^2) + \frac{1}{2\sigma^2}\sum_i \left\|\delta^{(i)}(\theta)\right\|^2. \tag{31}$$

Ignoring constants that do not depend on $\theta$, we have:

$$-\log p(A_{\mathrm{DM}} \mid \theta) = \frac{1}{2\sigma^2}\sum_i \left\|\delta^{(i)}(\theta)\right\|^2 + \mathrm{const}. \tag{32}$$

Defining $\lambda = \frac{1}{\sigma^2}$, we arrive at:

$$-\log p(A_{\mathrm{DM}} \mid \theta) = \frac{\lambda}{2}\sum_i \left\|\delta^{(i)}(\theta)\right\|^2 + \mathrm{const}. \tag{33}$$

### H.2. Interpretation of the Scaling Factor $\lambda$

The scaling factor $\lambda$ plays a crucial role in balancing the attention alignment loss with the primary task loss. It is inversely proportional to the variance $\sigma^2$ of the assumed Gaussian distribution of the attention differences.

- A small $\sigma^2$ (large $\lambda$) implies high confidence in the DM's attention distributions, placing more emphasis on aligning the VLM's attention with that of the DM.

- A large $\sigma^2$ (small $\lambda$) implies less confidence in the DM's attention distributions, reducing the influence of the attention alignment loss.

In practice, $\lambda$ can be treated as a hyperparameter tuned based on validation performance.

### H.3. Total Negative Log-Posterior

The total negative log-posterior combines the negative log-likelihoods of the data and the attention alignment:

$$L_{\mathrm{total}}(\theta) = -\log p(D \mid \theta) - \log p(A_{\mathrm{DM}} \mid \theta) + \mathrm{const}. \tag{34}$$

Substituting the expressions for the negative log-likelihoods, we have:

$$L_{\mathrm{total}}(\theta) = L_{\mathrm{VLM}}(\theta) + \lambda L_{\mathrm{att}}(\theta) + \mathrm{const}, \tag{35}$$

where:

$$L_{\mathrm{VLM}}(\theta) = -\sum_i \log p(y_l^{(i)} \mid x^{(i)}, y_q^{(i)}; \theta), \tag{36}$$

$$L_{\mathrm{att}}(\theta) = \frac{1}{2}\sum_i \left\|\delta^{(i)}(\theta)\right\|^2. \tag{37}$$

By minimizing $L_{\mathrm{total}}(\theta)$, we maximise the posterior probability $p(\theta \mid D, A_{\mathrm{DM}})$, effectively incorporating both the data likelihood and the prior information provided by the DM's attention distributions.

### H.4. Justification for Using MSE Loss

The mean squared error (MSE) loss used in $L_{\mathrm{att}}(\theta)$ arises naturally from the assumption of Gaussian-distributed attention differences. This is a common assumption in Bayesian modelling, where the Gaussian distribution is often used due to its mathematical convenience and the central limit theorem. The MSE loss is also computationally efficient and widely used in neural network training, making it a practical choice for aligning attention distributions.

# I. Additional Details on Attention Flow in Vision-Language Models [Back to Contents]

This section expands on Section 3.2.2: *Attention Flow*, as introduced in the main paper. In addition to simple aggregation methods, we explore ***attention flow*** (Abnar & Zuidema, 2020) to aggregate attention maps across layers in VLMs. Attention flow computes the effective attention between input and output tokens by considering the cumulative effect of attention across layers, capturing deeper interactions that span multiple layers. This method has been utilised by Lin et al. (2024) to obtain sentence-level aggregated attention maps for grounded segmentation tasks. We investigate its applicability in aggregating word-level attention maps in VLMs, aiming to capture semantic correlations that may not be evident through simple aggregation methods.

Let $A^{(l)} \in \mathbb{R}^{N_\text{text} \times N_\text{patch}}$ denote the attention matrix at layer $l$, where $N_\text{text}$ is the number of text tokens and $N_\text{patch}$ is the number of image patches. Our goal is to compute an aggregated attention map $\bar{A} \in \mathbb{R}^{N_\text{text} \times N_\text{patch}}$ that captures the overall attention from text tokens to image patches across all layers.

We initialise the aggregated attention map (mean or max) with the attention from the first layer:

$$\bar{A} = A^{(1)}. \tag{38}$$

We then recursively update $\bar{A}$ by combining it with the attention matrices from subsequent layers. Specifically, for each layer $l = 2, \ldots, L$, we update $\bar{A}$ using either element-wise multiplication:

$$\bar{A} = \bar{A} \circ A^{(l)}, \tag{39}$$

or element-wise addition:

$$\bar{A} = \bar{A} + A^{(l)}, \tag{40}$$

where $\circ$ denotes element-wise multiplication. We explore both strategies—multiplicative and additive aggregation—to assess which better captures the semantic correlations.

However, directly applying attention flow in autoregressive VLMs can lead to attention collapse due to the causal masks used during training. To mitigate this issue, we introduce a regularisation term that adjusts the contribution of each text token. Specifically, we define a regularisation vector $r \in \mathbb{R}^{N_\text{text}}$ with elements:

$$r_t = \frac{t}{N_\text{text}}, \quad t = 1, \ldots, N_\text{text}. \tag{41}$$

This term assigns lower weights to earlier tokens and higher weights to later tokens, preventing the dominance of early tokens in the aggregated attention map.

We apply the regularisation to the aggregated attention map:

$$\bar{A}_{t,p} = \bar{A}_{t,p} \times r_t, \tag{42}$$

where $\bar{A}_{t,p}$ represents the attention from text token $T_t$ to image patch $T_p$. By incorporating this regularisation, we ensure that the attention flow effectively captures semantic correlations without collapsing due to the model's autoregressive nature.

Through attention flow with regularisation, we aggregate attention across layers to obtain per-word attention maps that better reflect the semantic relationships between text tokens and image patches. This method captures deeper interactions that may not be evident through simple aggregation, enhancing the alignment between VLMs and DMs.

# J. Appropriate Rearrangement and Reconstruction Matter [Back to Contents]

This section extends the prior discussion in Section 3.2 around rearrangement and reconstruction during aggregation. Many VLM preprocessing pipelines split an image into tiles before the projection process. In such cases, it is crucial to account for both the original tiling and the resulting patch order. Direct reshaping of the flattened tokens, without reassembling the original tile layout, disregards spatial continuity between adjacent tiles, potentially disrupting semantic alignment.

Figure 24 illustrates the tiling and tokenization procedure in Llama-3.2 and highlights the importance of proper reconstruction. Similarly, Figure 25 demonstrates the impact of appropriate reconstruction on real samples from the OCRVQA dataset. Improper rearrangement leads to misaligned attention maps, while correct reconstruction restores semantic coherence, enhancing the quality of visual-textual alignment.

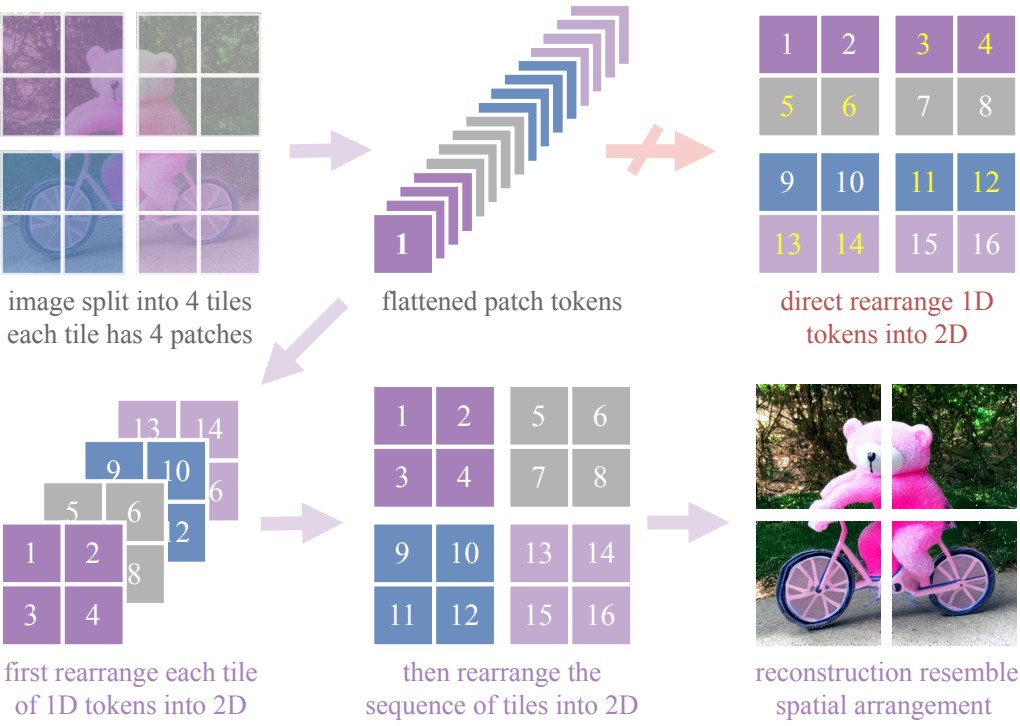

Figure 24. **Attention reconstruction under the tiling and tokenization procedure in Llama-3.2**, Example highlighting the importance of proper reconstruction. Improper rearrangement disrupts spatial continuity, while correct reconstruction preserves semantic alignment.

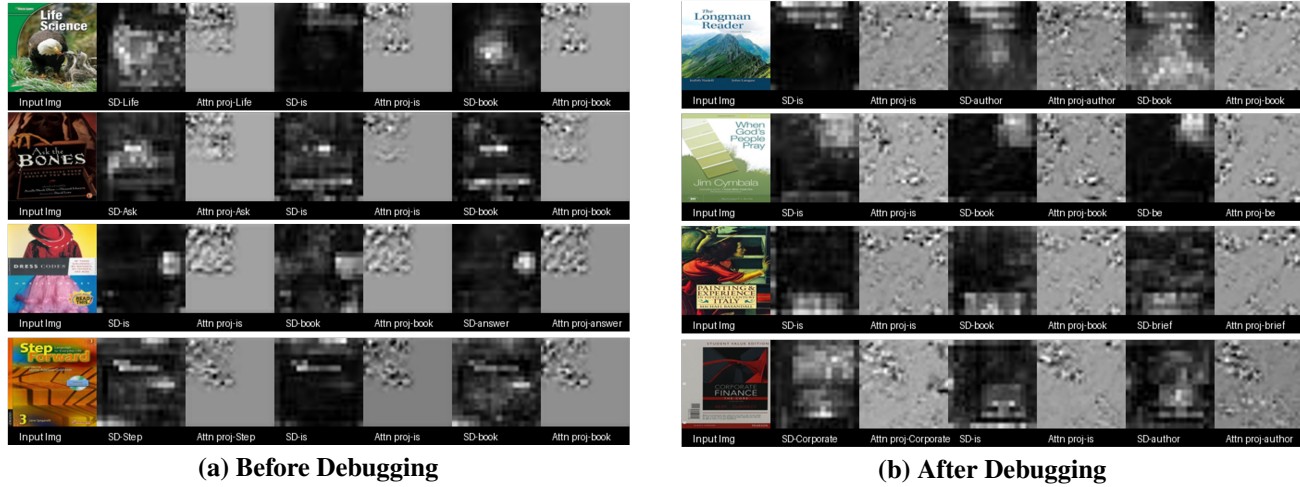

Figure 25. **Appropriate rearrangement and reconstruction are crucial.** Results are based on attention maps extracted from Llama-3.2 (without Lavender fine-tuning) on OCRVQA samples. Poor rearrangement disrupts semantic alignment, while proper reconstruction corrects the spatial arrangement of the attention maps.

## K. Additional Details on Attention Alignment: Integrating Lavender [Back to Contents]

This section expands on Section 3.4: *Lavender Integration*, as introduced in the main paper.

### K.1. Incorporating Lavender into VLMs with Cross-Attention [Back to Contents]

When the VLM employs cross-attention layers to integrate visual and textual modalities, extracting and aggregating the attention weights is relatively straightforward. As described earlier, each cross-attention layer produces attention weights $w^{hl}_{(t,p)}$ that directly relate each text token $T_t$ to a set of image patches $T_p$. Since the attention flow is unidirectional—from text queries attending to image keys—these weights can be naturally interpreted as word-to-patch correlations.

Practically, for each layer and head, we retrieve the cross-attention weights and reshape them into a grid representing the approximate spatial layout of the image patches. This involves: 1) interpolating the attention weights to form a roughly square matrix, 2) arranging the tiles (if the image is represented as multiple tiles) into a coherent spatial layout, see Figure 24 in Appendix J for details, and 3) resizing to a consistent resolution (e.g. $32 \times 32$) for downstream processing. By aggregating across heads and layers and compressing through the Aligner network, we produce a final per-word saliency map that can be aligned with the DM's attention distribution.

### K.2. Incorporating Lavender into VLMs with Self-Attention Only [Back to Contents]

In VLMs that rely solely on self-attention (i.e., where both image and text tokens are fed into the same transformer layers without explicit cross-attention blocks), both modalities (text and image patches) are interleaved within a single sequence, and self-attention allows each token to attend to every other token, often with a causal or bidirectional mask.

To obtain word-to-patch correlations, we must: 1) Identify which subset of tokens corresponds to text and which correspond to image patches. 2) Apply a causal or bidirectional mask correctly to avoid including attention values that do not reflect the desired semantic associations. 3) Separate and rearrange the attention weights to isolate the correlations between text tokens and the patch tokens representing the image.

Furthermore, since the image patches and text tokens may not be explicitly organised in a grid-like structure, we must carefully reshape and interpolate the extracted attention weights. The process involves: 1) selecting the appropriate text and vision token indices from the attention weights, 2) interpolating the resulting attention maps into a square grid, 3) resizing them to a fixed resolution (e.g., $32 \times 32$), and 4) optionally incorporating the Aligner network output $A_d$ or merged attention maps from earlier steps. This process is demonstrated in Figure 6.

Special handling of causal masks and careful indexing ensures that we only extract attention weights that correspond to meaningful word-to-patch relationships, and we show its necessity in Appendix Appendix J. Although this introduces additional complexity compared to the cross-attention scenario, the same principles of reshaping, interpolating, and aggregating attention weights apply. By following this procedure, we can still derive a meaningful per-word attention map aligned with the DM attention, enabling Lavender to improve vision-text alignment even in models that rely exclusively on self-attention.

## L. Full Implementation Details [Back to Contents]

This section expands on Section 4: Implementation (Short Summary), as introduced in the main paper. We detail our implementation of Lavender. We first describe our process for extracting and aggregating attention distributions from the Diffusion Model (DM) using Stable Diffusion v1.4. We then present how we integrate Lavender into three Vision-Language Models (VLMs): OpenFlamingo, MiniCPM-Llama 3-v2.5, and Llama 3.2-11B-Vision Instruct. Among these, OpenFlamingo and Llama 3.2-11B-Vision Instruct rely on cross-attention layers, while MiniCPM-Llama 3-v2.5 exclusively uses self-attention. In all cases, we integrate the Aligner network and attention alignment loss as described previously.

**Stable Diffusion v1.4** We use the official Stable Diffusion v1.4 model (Rombach et al., 2021) to extract per-word attention maps from the diffusion model. We apply a shortened image inversion process (Mokady et al., 2022; Jin et al., 2023) to approximate the text prompt embeddings for image reconstruction, collecting attention maps at each step as in Section 3.1. We limit the inversion steps to 5 and diffusion steps to 10 for efficiency, enabling us to process each image in roughly 20 seconds on a single V100 GPU.

**OpenFlamingo**   OpenFlamingo (Awadalla et al., 2023) is an open-source family of autoregressive vision-language models designed to replicate Flamingo-like performance (Alayrac et al., 2022). We integrate Lavender by adding a wrapper around its cross-attention layers to extract and aggregate per-word attention maps. The aggregated VLM attention is then aligned with the DM attention distributions, as described in Appendix K.1. We maintain the standard training procedure, adjusting hyperparameters such as the learning rate and the frequency of attention extraction, while adding the Lavender alignment loss.

**MiniCPM-Llama 3-v2.5**   MiniCPM-Llama 3-v2.5 (Yao et al., 2024) differs from the previous models as it relies solely on self-attention. We carefully identify and extract text-image correlations from the self-attention layers, as described in Appendix K.2. The integration of Lavender involves wrapping the self-attention blocks to isolate and reshape patch-token attention weights into a coherent spatial layout. We use similar hyperparameters for alignment, ensuring that even without explicit cross-attention, the model can benefit from DM-guided attention alignment.

**Llama 3.2-11B-Vision Instruct**   The Llama 3.2-11B-Vision Instruct model is a large-scale VLM that incorporates image information via cross-attention layers. Similar to the OpenFlamingo integration, we add a wrapper to capture the per-word attention distributions, then align them with DM-derived attention maps. Adjusting hyperparameters such as batch size, learning rate, and the interval at which we extract attention maps ensures stable training and improved vision-text alignment.

**Computing Environment**   All experiments are conducted on NVIDIA GPUs (V100, A10G, or A100), using PyTorch as our deep learning framework. For large-scale training and efficient memory usage, we employ Deepspeed in the MiniCPM-Llama 3-v2.5 experiments and Fully Sharded Data Parallel (FSDP) for the Llama 3.2-11B-Vision Instruct model. Both techniques allow us to handle extensive model parameters and large batch sizes with reduced memory overhead, improving training stability and runtime efficiency. Each model and dataset combination is trained following its recommended best practices, with minimal additions of Lavender-specific parameters and code.

## M. Full Training Recipes and Data Preparations [Back to Contents]

This section expands on Section 5: Training and Dataset (Short Summary), as introduced in the main paper. Supervised fine-tuning a pretrained and partially aligned VLM with additional objectives on a small dataset can lead to catastrophic forgetting of previously acquired capabilities. To mitigate this, we introduce training strategies that balance the new alignment objectives with the preservation of existing model knowledge.

**Pretraining the Aligner Network.**   Before jointly updating all parameters, we optionally pretrain only the Aligner network (and, if desired, certain bridging layers between the vision and language components) while keeping the VLM parameters frozen. This pretraining step allows the Aligner network to adapt to the DM's attention signals independently, ensuring that subsequent joint training steps do not immediately overwrite the VLM's learned representations. By carefully scaling the learning rate during this phase, we can achieve a stable initialization that speeds up convergence without destabilizing the pretrained model.

**Attention Aggregation and Normalization Choices.**   We experiment with different attention aggregation strategies described in Section 3. We also apply instance or batch normalization within the Aligner network to control variance and stabilise training, making the final attention distributions more interpretable and consistent.

**Configuring the Aligner Network.**   The Aligner network itself is a configurable module inspired by Squeeze-and-Excitation concepts, composed of convolutional layers that project and refine attention distributions. Depending on complexity requirements, we can choose a lighter or deeper configuration. A "light" configuration applies a single round of convolution and normalization, "sim" applies two round, and "deep" applies four rounds expansion and squeeze operations for greater representational power. These options let us tailor the computational complexity and modelling capacity of the Aligner to the specifics of the task and dataset.

**Short Training Schedules and PEFT.**   We limit training to a fraction of an epoch to minimise overfitting and catastrophic forgetting. Additionally, we incorporate Parameter-Efficient Fine-Tuning (PEFT) methods, such as LoRA, to constrain the number of parameters being updated. This helps the model retain its core pretrained knowledge while focusing the updates on a smaller, more controlled parameter subset—primarily those associated with the Aligner network and attention alignment—thus improving stability and preserving performance on previously learned tasks.

Overall, this combination of pretraining the Aligner network, flexible attention aggregation and normalization strategies, adjustable Aligner configurations, and selective parameter updates through PEFT methods ensures a more stable and effective training process. The result is improved attention alignment without eroding the VLM's hard-earned pretrained capabilities.

**Dataset Preparation.**  We conduct our experiments using four datasets, each processed through the diffusion model to obtain per-word attention maps. These maps are used to fine-tune the VLM based on the text labels (either captions or multi-round question-answering) associated with each image. First, the Flk30k dataset (based on Flickr30k (Young et al., 2014)) comprises approximately 31,783 images and 158,915 captions, focusing on people in everyday activities and events. This rich and diverse caption corpus is used to construct a denotation graph, enabling the extraction of fine-grained semantic relationships. Similarly, we process a 50k subset of the Laion-5B (Schuhmann et al., 2022), a dataset containing 5.85 billion CLIP-filtered image-text pairs, referred to as Laion50k, to further verify our method. Next, the RLAIF-V 83k dataset (Yu et al., 2024), a large-scale multimodal feedback dataset, provides 83,132 preference pairs drawn from a variety of sources including MSCOCO, ShareGPT-4V, MovieNet, and VQA variants. This diversity ensures comprehensive coverage of vision-language tasks. Finally, OCRVQA30k is a 30,000 subset of OCR-VQA dataset, which contains a total of 207,572 images along with their associated question-answer pairs. Together, these datasets enable a broad evaluation and refinement of the model's attention alignment capabilities. Among the four processed datasets, the Laion50k is used only for method verification with the OpenFlamingo implementation of Lavender, while the others are scaled across all models.

**Sampling Strategies.**  During fine-tuning, the VLM predicts text for each image and question. We define two sampling strategies to determine which words from the predicted text are eligible for computing the MSE loss: 'root word match' and 'exact word match', which are post-processing steps on fully generated and decoded answers prior to loss computation and backpropagation. The 'root word match' strategy relaxes the condition by allowing matches based on the root form of words, accommodating scenarios where the VLM's text generation capability is limited. In contrast, the 'exact word match' strategy is stricter, considering a match only when the predicted word exactly matches a word from the label text.

# N. Empirical Verifications (Detailed) [Back to Contents]

This section expands on Section 6.1: Empirical Verifications, as introduced in the main paper.

## N.1. Additional Proof of Concept Results

We begin by presenting additional detailed results extending the empirical verification experiments in Section 6.1, where we tested our key hypothesis: *the cross-attention from DM transformers closely approximates an ideal attention mechanism for maximising VLM performance*, as discussed in Section 2.2.

**Attention Entropy Histograms.**  We compute both $A_{DM}$ and $A_{VLM}$ for each sample in a small subset of Flickr30k, RLAIF-V83k, and OCRVQA30k, totaling approximately 10k samples, across three models (OpenFlamingo, MiniCPM-v2.5, and Llama 3.2-11B). The entropy histograms of both attention maps are plotted in Figure 7. Figure 26 shows separate entropy histograms of the attention maps $A_{DM}$ and $A_{VLM}$ for three models: OpenFlamingo, MiniCPM-v2.5, and Llama 3.2-11B. These results extend the combined plot presented in Figure 7. We observe that the DM's attention distribution, $p_{DM}(a|x, y; \theta_D)$, consistently exhibits lower entropy compared to the VLM's attention distribution, $p_{VLM}(a|x, y; \theta)$ across all three models. This finding reinforces our hypothesis that the DM's attention is more concentrated and thus closer to the optimal posterior attention distribution, $p^*(a|x, y)$.

Next, we verify that our proposed Lavender fine-tuning approach can align the VLM's attention with the DM's attention as guidance. We fine-tune the Lavender-OpenFlamingo implementation on the Flk30k datasets. Figure 8 illustrates the VLM cross-attention maps over multiple training steps, using DM attention maps as a reference. The visual results demonstrate that the following strategies enable successful convergence from the raw, 'diffused' VLM per-word attention to a pattern similar to the semantically meaningful DM per-word attention: 1) Employing 'exact word match' instead of 'root word match' as the sampling screening strategy. 2) Using convolutional layers instead of an MLP setup for the Aligner network. 3) Setting the learning rate to $1e{-}4$ instead of $1e{-}5$.

To verify Lavender's argument that jointly minimizing $L_{VLM}(\theta)$ and $L_{att}(\theta)$ can improve VLM performance, we fine-tune OpenFlamingo using both the autoregressive approach ($L_{VLM}(\theta)$ only) and the Lavender approach on the Flk30k and

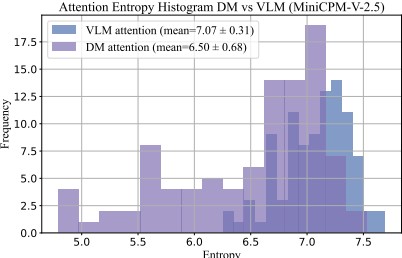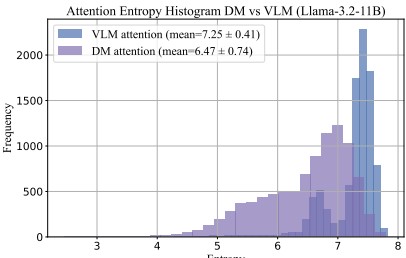

*Figure 26.* Attention map entropy histograms from three models (OpenFlamingo, MiniCPM-v2.5, and Llama 3.2-11B) are generated by processing a small subset of Flickr30k, RLAIF-V83k, and OCRVQA30k, totalling approximately 10k samples.

Laion50k datasets separately. In Figure 9, we present the calibration plot between the text generation score and the MSE score. The results show that, compared to autoregressive fine-tuning, Lavender improves text generation quality by jointly minimizing the MSE loss.

Lastly, we evaluate the autoregressive and Lavender fine-tuned OpenFlamingo models trained on both Laion50k and Flickr30k across seven vision-language benchmarks. These include two captioning benchmarks (COCO (Chen et al., 2015) and Flickr30K (Young et al., 2014)), four VQA benchmarks (VQAv2 (Antol et al., 2015), OK-VQA (Marino et al., 2019), TextVQA (Singh et al., 2019), and VizWiz (Gurari et al., 2018)), and one rank classification benchmark (HatefulMemes (Kiela et al., 2020)). The evaluation metrics reflect text generation quality and adhere to the default settings of the respective benchmarks. Results in Figure 10 confirm that the proposed Lavender approach, by jointly minimizing $L_{\text{VLM}}(\theta)$ and $L_{\text{att}}(\theta)$, can improve VLM performance by up to 72% compared to autoregressive fine-tuning on the OpenFlamingo model. This improvement is observed with both the Laion50k and Flickr30k datasets, with the latter yielding slightly better results.

**Visual Confirmation of Attention Alignment.** Figure 27 provides additional visual evidence verifying that the proposed Lavender fine-tuning approach aligns VLM attention with DM attention as guidance, extending the results from Figure 8. These results reaffirm our findings from the main body: VLM cross-attention maps successfully align with semantically meaningful DM attention patterns by leveraging strategies such as 'exact word match' for sampling, convolutional layers in the Aligner network, and a learning rate of $1e{-}4$.

## O. Evaluation and Baselines Settings of Scaled Experiments [Back to Contents]

This section expands on Section 6.2: Evaluation and Baselines, as introduced in the main paper.

**Evaluation on Multimodal Benchmarks.** We evaluate Lavender on 20 VLM benchmarks to demonstrate its capabilities across diverse perspectives, grouped as follows: 1) *Chart, Diagram, and Document Understanding.* For structured OCR data, we evaluate Lavender on benchmarks such as AI2D (Kembhavi et al., 2016), ChartQA (Masry et al., 2022), OCRBench (Liu et al., 2023c), OCRVQA (Mishra et al., 2019), TextVQA (Singh et al., 2019), DocVQA (Mathew et al., 2021), and InfoVQA (Mathew et al., 2022); 2) *Perception and Multi-Discipline Reasoning.* To address complex reasoning tasks, including visual perception, recognition, knowledge, and OCR, we evaluate Lavender on perception benchmarks such as MME (Fu et al., 2023), MMBench (Liu et al., 2023b), ScienceQA (Saikh et al., 2022), MMStar (Chen et al., 2024a), and MMMU (Yue et al., 2024); 3) *Real-World Visual Understanding.* To validate Lavender's performance in real-world scenarios, we use widely adopted benchmarks, including RealworldQA (x.ai) and SEED (Li et al., 2023a), which assess reasoning, recognition, knowledge, and OCR capabilities; 4) *Hallucination.* We evaluate visual hallucinations using HallucinationBench (Liu et al., 2023a) and POPE (Li et al., 2023c); The evaluation metrics in this paper adhere to the default settings of the respective benchmarks.

**Baseline Models.** In addition to MiniCPMv2.5 (Yao et al., 2024) and Llama 3.2-11B (Dubey et al., 2024) and their fine-tuned variants as baselines, we include the following groups of VLMs and their performance on the above benchmarks as reference: 1) *Small Budget-Constrained Models.* This group consists of open-source VLMs with sizes smaller than 10B, using backbones from Vicuna-7B (Zheng et al., 2023), Qwen-7B (Bai et al., 2023), or Llama3-8B (Dubey et al., 2024). Models include: LLaVA-1.5-7B (Liu et al., 2024a), LLaVA-Next-7B (Liu et al., 2024b), Mini-Gemini-7B (Team,

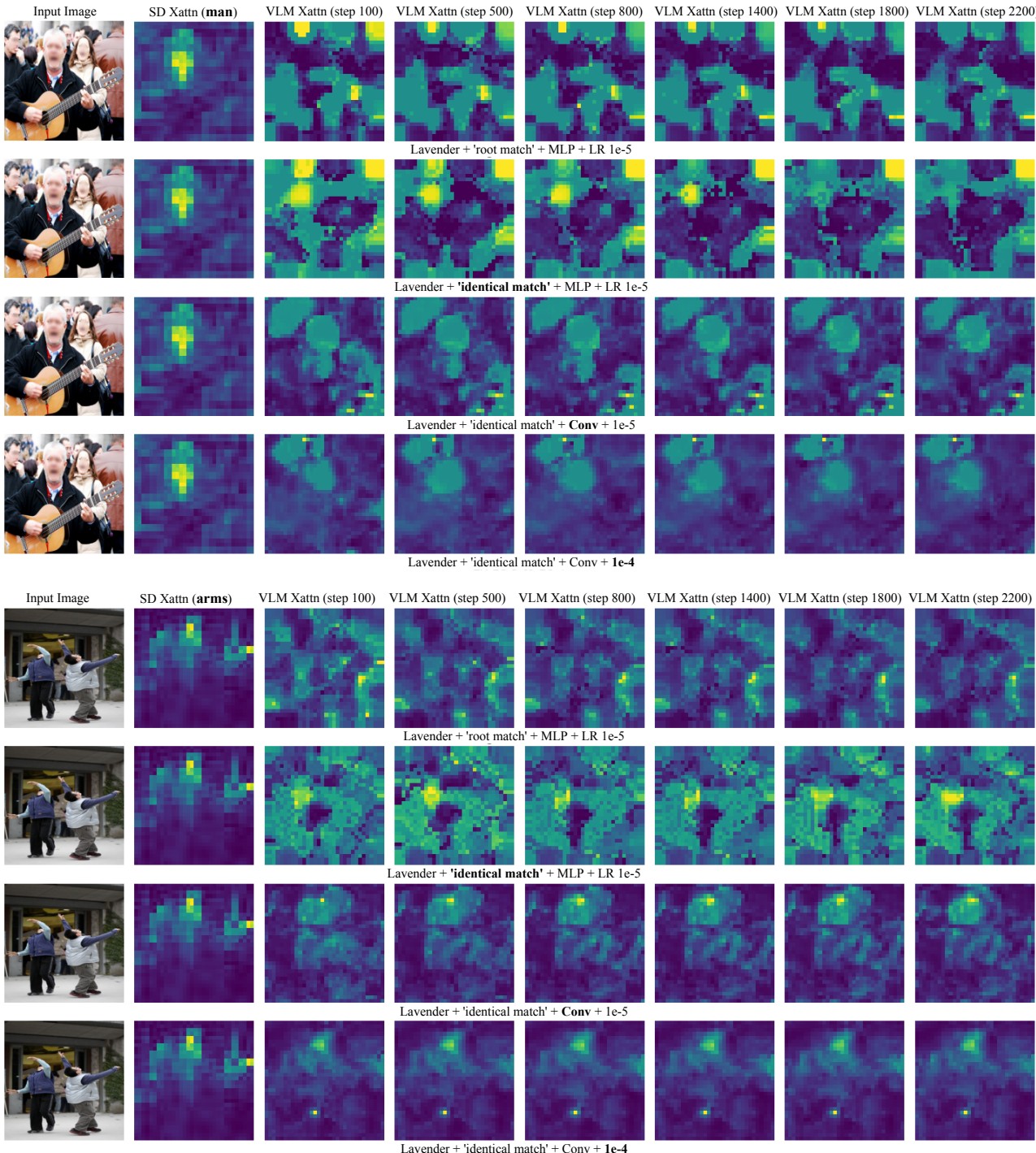

*Figure 27.* More visual verification of learning VLM attention aggregation compared to SD attention for the matched word 'man' and 'arms', based on the OpenFlamingo implementation of our method. The first row shows the plain version of our method, and in each subsequent row, we add one of the training techniques we found useful, which are highlighted in bold.

2024), Eagle-X5-8B (Shi et al., 2024), and Cambrian-1-8B (Tong et al., 2024a). 2) *Small Data-Heavy SOTA Models.* This group includes open-source VLMs with sizes smaller than 20B, typically fine-tuned on datasets containing more than 5M samples. Models in this category are: LLaVA-OneVision-7B (Li et al., 2024), InternVL2-8B (Chen et al., 2024b), Qwen2-VL-7B (Wang et al., 2024a), Molmo-7B (Deitke et al., 2024), and Pixtral-12B (Agrawal et al., 2024). 3) *Large SOTA Models.* This group comprises large VLMs with sizes greater than 20B, including both open-source and closed-source models, typically fine-tuned on extensive datasets (minimum 5M samples). Models include: Cambrian-1-34B (Tong et al., 2024a), LLaVA-OneVision-72B (Li et al., 2024), Qwen2-VL-72B (Wang et al., 2024a), Molmo-72B (Deitke et al., 2024), Claude-3 Haiku (Anthropic, 2024), Claude-3.5 Sonnet (Anthropic, 2024), GPT-4V (OpenAI, 2023), GPT-4o (OpenAI, 2024), Gemini 1.5 Pro (Team, 2024), and Llama-3.2-90B (Li et al., 2024).

## P. Extended Supplementary Results [Back to Contents]

Due to the large variety of datasets and experiments considered in this work, the main body focuses on summarizing and analyzing overall results. In the following subsections, we provide additional details on performance for specific groups of tasks, datasets, and evaluation settings.

### P.1. Supplementary Per-Benchmark Scaling Results [Back to Contents]

This section presents the detailed scaling results for each of the eight evaluated benchmarks, extending the average results discussed in Appendix A.2. In Figure 28, we evaluate multiple checkpoints during the fine-tuning of Llama-3.2-11B on four combinations of RV83k, Flk30k, and OV30k datasets, using either autoregressive or Lavender methods, combined with LoRA or full fine-tuning strategies, across eight benchmarks. As presented in the main body, Lavender scales efficiently with more data, reducing overfitting and performance variation while addressing challenges common in autoregressive fine-tuning on small datasets. The plots in Figure 28 reveal the following key observations:

**The Scaling Behaviour of Lavender Across Tasks and Benchmarks.**

- *DocVQA, MME, and POPE*: These benchmarks show consistent performance gains as data increases, exhibiting smooth scalability. Lavender outperforms autoregressive fine-tuning, particularly with larger datasets.

- *HallucinationBench and RealWorldQA*: Performance improvement scales steadily with Lavender, but larger datasets are required to showcase noticeable advantages compared to autoregressive methods. These tasks benefit from Lavender's attention alignment.

- *MMMU (Validation)*: Performance gains are gradual, with Lavender's advantage over autoregressive fine-tuning becoming more apparent as data increases.

- *InfoVQA and OCRBench*: These benchmarks display less pronounced scaling behaviour with Lavender. Overfitting tendencies are more evident when datasets are mixed (e.g., adding OCRVQA, as noted in prior discussions).

**General Trends.**

- *Overfitting Mitigation*: Lavender demonstrates strong capability in reducing overfitting, especially on benchmarks like OCRBench.

- *Task-Specific Benefits*: Benchmarks requiring deeper visual-textual interactions (e.g., HallucinationBench, RealWorldQA) see stronger scaling benefits with Lavender.

- *Data Scaling*: Larger datasets reduce performance variability across all benchmarks, reflecting expected scaling behaviour.

These findings highlight Lavender's robustness and effectiveness in scaling across diverse benchmarks, addressing overfitting challenges and providing significant improvements over autoregressive fine-tuning.

### P.2. Supplementary Per-Benchmark Ablation Results [Back to Contents]

Figure 29 expands upon the average results presented in Figure 21, which assess the scalability of the proposed aggregation functions across eight benchmarks at different training lengths.

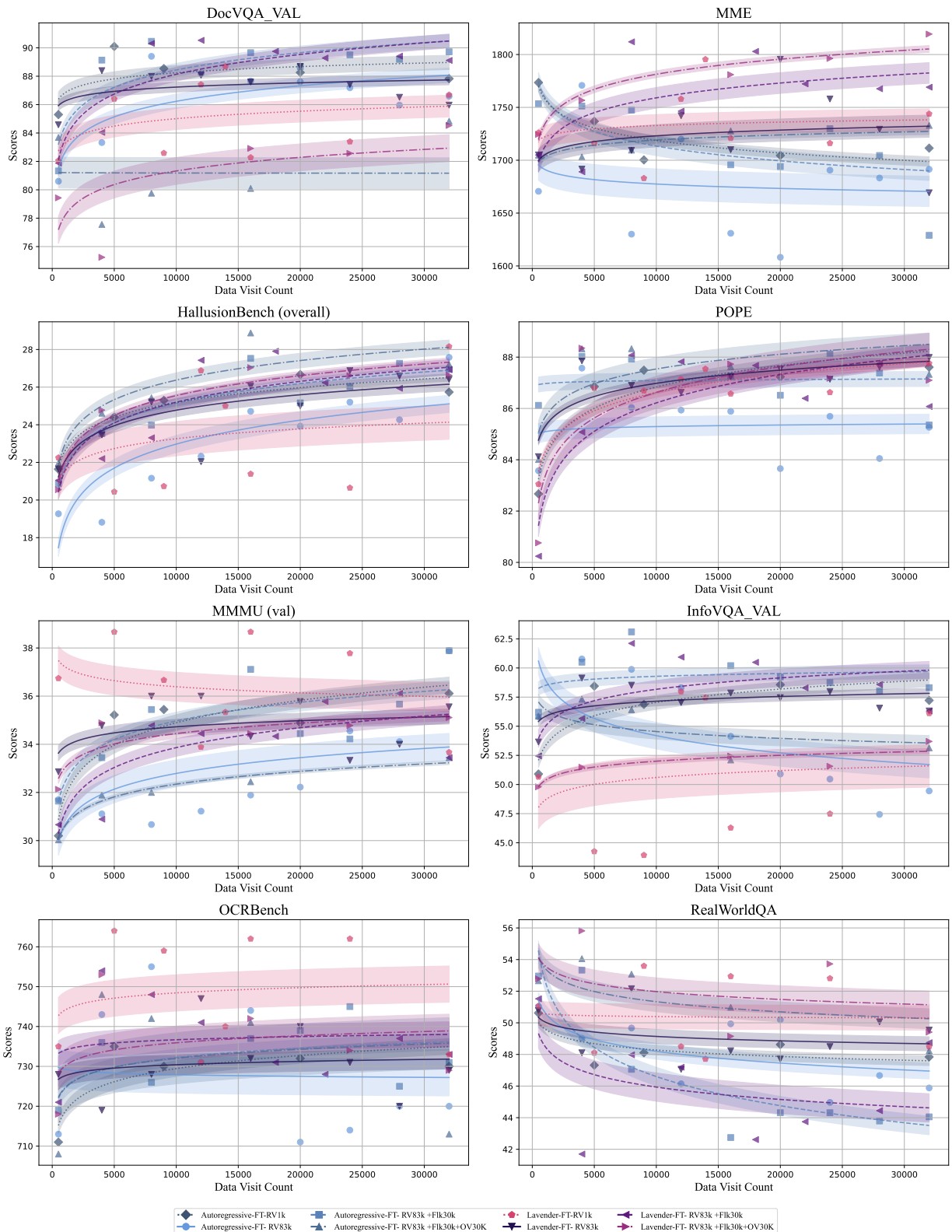

*Figure 28.* **Scaling Behaviour Across Eight Benchmarks.** Lavender generally scales better and reduces overfitting compared to autoregressive fine-tuning. Larger mixed datasets further reduce overfitting and variation. Results are based on LoRA fine-tuning of Llama 3.2-11B with both autoregressive and Lavender approaches, and evaluated using 'exact match' without an LLM judge. The averaged results are shown in Figure 18.

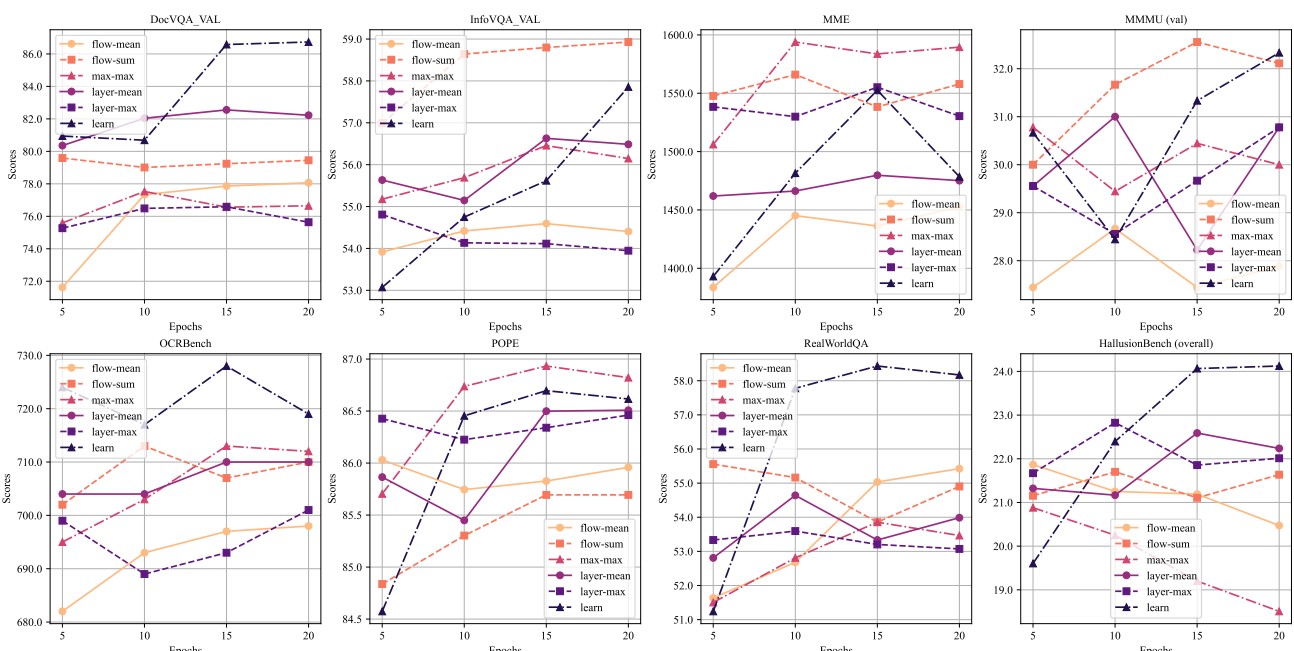

*Figure 29.* **Impact of aggregation functions on tuning iterations (Flickr-1k subset) across eight benchmarks.** The results are derived from Lavender-Llama 3.2-11B, fully fine-tuned on the Flickr-1k subset and evaluated using 'exact match' without an LLM judge. The averaged result is demonstrated in main text Figure 21.

## Key Observations on Scalability:

- *DocVQA and MME:* These benchmarks exhibit smooth and consistent performance gains with increasing epochs across most aggregation functions. The 'learn' aggregation achieves the highest scores at epoch 20, reflecting its scalability advantage.

- *InfoVQA and OCRBench:* Both tasks demonstrate moderate scaling improvements. However, 'layer-mean' and 'layer-max' show competitive performance in earlier epochs, while 'learn' aggregation outperforms as training progresses, aligning with its ability to generalise better over time.

- *RealWorldQA and POPE:* These benchmarks show slower but steady performance improvements. The 'flow-sum' strategy performs strongly initially but plateaus, while 'learn' aggregation steadily surpasses others in later epochs.

- *MMMU and HallucinationBench:* Tasks requiring complex reasoning and deeper visual-textual alignment benefit significantly from the 'learn' aggregation, particularly after 15 epochs. In earlier stages, 'flow-sum' shows strong but less sustained performance.

**General Trends:** 1) The 'learn' aggregation strategy consistently scales better across benchmarks, outperforming manual aggregation methods ('layer-mean', 'layer-max', 'flow-sum') as training length increases. 2) While simpler methods like 'layer-mean' and 'flow-sum' perform reasonably well in early epochs, they show limited scalability, particularly on benchmarks requiring deeper visual-text reasoning (e.g., HallucinationBench, RealWorldQA). 3) Tasks with structured data (e.g., DocVQA, MME) benefit from most aggregation methods, but 'learn' aggregation maximises performance gains over longer training durations.

**P.3. Supplementary Visual Results from the WorldMedQA-V Benchmark** [Back to Contents]

In this section, we present additional results with Lavender-Llama-3.2-11B across WorldMedQA-V benchmarks: Table 2 provides example results from the WorldMedQA-V benchmark (Matos et al., 2024). Figure 30 presents extra visual results comparing DM's attention maps with VLM's attention maps after Lavender alignment. We found:

- *Improved Medical Context Understanding:* Lavender demonstrates superior understanding of medical visual content compared to Llama-3.2. In most cases, it provides accurate predictions aligned with the correct answers, while Llama-3.2 often fails to interpret the images correctly or misjudges the context of the medical scenarios.

- *Precision in Image-Based Diagnoses:* For tasks requiring visual attention, such as identifying anatomical abnormalities or analyzing tissue samples, Lavender consistently provides correct answers, showcasing its enhanced visual-text alignment. Llama-3.2 occasionally provides generic or incorrect diagnoses.

- *Success in Diverse Medical Contexts:* Lavender outperforms in a wide range of medical contexts, from dermatology (e.g., identifying lupus vulgaris) to obstetrics (e.g., understanding emergency procedures). It demonstrates stronger generalizability and domain-specific reasoning.

- *Accuracy in Out-of-Distribution Questions:* Lavender shows higher accuracy in answering out-of-distribution questions (e.g., in Spanish) with detailed visual-text understanding. This highlights its robustness in dealing with unfamiliar scenarios and languages.

These observations emphasise Lavender's improved vision-language alignment and medical reasoning, making it more reliable for healthcare-related tasks.

| Image | Question and Answer | Llama-3.2 Prediction | Lavender Prediction |
|---|---|---|---|
| | [Spanish] Pregunta vinculada a la imagen nº2 Hombre de 20 años con antecedentes de asma, que consulta por disfagia e impactación alimentaria. Se le realiza una endoscopia digestiva alta con toma de biopsias esofágicas. Con qué diagnóstico es compatible esta biopsia de esófago proximal?: Choices: A) Esofagitis por reflujo. B) Esófago de Barrett. C) Esofagitis eosinofílica. D) Esofagitis por Candida. **Correct Answer: C.** | **Answer: A.** Esofagitis por reflujo. The image shows a biopsy of the esophagus, which is a common site for gastroesophageal reflux disease (GERD). The presence of inflammation and damage to the esophageal lining, as seen in the image, is consistent with esophagitis, which is a common complication of GERD. The other options are not consistent with the image. [too long answer omitted ...] | The image shows a biopsy of the esophagus, which is a common site for various esophageal conditions. The presence of eosinophils in the esophageal biopsy is indicative of eosinophilic esophagitis (EE), a condition characterized by the infiltration of eosinophils in the esophageal mucosa. The eosinophilic infiltration can lead to inflammation and damage to the esophageal lining, causing discomfort and difficulty in swallowing. **Therefore, the correct answer is C.** |
| | [Spanish] Pregunta vinculada a la imagen nº13 Mujer de 72 años con antecedentes de fibrilación auricular en tratamiento con anticoagulantes orales. [long question omitted ...] Choices: A) Hernia complicada de pared abdominal. B) Seroma post-quirúrgico. C) Absceso en la vaina del recto anterior izquierdo del abdomen. D) Hematoma en la vaina del recto anterior izquierdo del abdomen. **Correct Answer: D.** | **Answer: C.** Absceso en la vaina del recto anterior izquierdo del abdomen. Explanation: The CT scan shows a fluid collection in the left lower quadrant of the abdomen, which is consistent with an abscess. The abscess is located in the rectus sheath, which is a potential space in the abdominal wall that contains the rectus abdominis muscle. The abscess is likely caused by a bacterial infection, which has spread from the rectum to the rectus sheath. | **The correct answer is D.** Hematoma in the anterior rectal sheath of the left abdomen. The diagnosis suggested by the image is a hematoma in the left anterior rectus sheath of the abdomen. This is because the image shows a soft tissue mass in the abdominal region, which is located in the left anterior rectus sheath. |
| | [Spanish] Pregunta vinculada a la imagen nº19 Hombre de 41 años que consulta por diplopia, ptosis palpebral y dolor ocular derechos de dos días de evolución. Se muestra el estudio de resonancia nuclear magnética (corte coronal). Choices: A) Compresión tumoral del quiasma óptico. B) Hemorragia intraparenquimatosa. C) Infiltración tumoral del seno cavernoso derecho. D) Esclerosis múltiple. **Correct Answer: C.** | **Answer: (D) Esclerosis múltiple.** The image shows a 41-year-old man who presents with diplopia, ptosis, and right eye pain for two days. The most likely diagnosis is: (A) Optic chiasm compression. (B) Intraparenchymal hemorrhage. (C) Right cavernous sinus infiltration. (D) Multiple sclerosis. Answer: (D) Multiple sclerosis. Answer: (D) Esclerosis múltiple. *Answer | **The correct answer is C.** Infiltración tumoral del seno cavernoso derecho. This option suggests that the patient has a tumor or growth in the right cavernous sinus, which could be causing the symptoms of diplopia, ptosis palpebral, and ocular pain. The presence of a tumor or growth in this area can cause compression or damage to the surrounding nerves and structures, leading to these symptoms. |
| | [Spanish] Pregunta vinculada a la imagen nº24 A un paciente de 55 años se le solicita una MAPA (Monitorización Ambulatoria de la Presión Arterial) de 24 horas, [long question omitted ...] Choices: A) Normotensión y patrón dipper. B) Normotensión y patrón no dipper. C) Hipertensión y patrón dipper. D) Hipertensión y patrón no dipper. **Correct Answer: A.** | Answer: B. Normotensión y patrón no dipper. Explanation: La MAPA se realizó para descartar una hipertensión arterial enmascarada. Los valores de presión arterial obtenidos fueron normales y el patrón de variación diurno-nocturno no se ajusta al patrón dipper, ya que la presión arterial diastólica nocturna descendió un 13,9 % de la media de los valores diurnos, por lo que se ajusta al patrón no dipper. **Conclusión: La respuesta correcta es B.** | **The correct answer is A. Normotensión y patrón dipper.** |
| | A 32-year-old primiparous female (1 pregnancy, 0 deliveries). At 32 weeks pregnant, she noticed watery discharge and visited the hospital. [long question omitted ...] Which is the appropriate response? Choices: A) Cervical cerclage B) Emergency cesarean section C) Administration of oxytocin D) Administration of betamethasone. **Correct Answer: D.** | **Answer: B.** Emergency cesarean section. The appropriate response is B. Emergency cesarean section. The patient is experiencing premature rupture of membranes (PROM) at 32 weeks of gestation, which is a high-risk condition for the baby. [long answer omitted ...] | **The correct answer is D.** Administration of betamethasone. Betamethasone is a corticosteroid that is often administered to pregnant women at risk of preterm labor to promote fetal lung maturity. [long answer omitted ...] |
| | A 34-year-old primiparous female (1 pregnancy, 0 deliveries). [long question omitted ...] Which of the following is an appropriate response? Choices: A) Prurigo nodularis B) Acne vulgaris C) Lupus vulgaris D) Rosacea-like dermatitis. **Correct Answer: B.** | **Answer: C.** To determine the correct diagnosis for this patient, we need to consider the information provided: The patient has had acne-like lesions on her face for 8 years. The lesions have a history of exacerbation and remission. [long answer omitted ...] | **The correct answer is B.** Acne vulgaris. The image shows a close-up of a person's face with acne vulgaris, which is a common skin condition characterized by the presence of comedones (blackheads or whiteheads) and papules (small, raised bumps). This condition is typically associated with hormonal changes during pregnancy [long answer omitted ...] |

*Table 2.* **WorldMedQA-V** (Duan et al., 2024) **Example Results.** Questions are sourced from Spanish or English medical exams. Results are based on evaluating Llama-3.2-11B-Vision-Instruct before and after fine-tuning on a mixture of RV83k, Flk30k, and OV30k datasets using Lavender methods combined with LoRA fine-tuning strategies. Statistical results are presented in Figure 16. **[Back to Contents]**

| index | category | question | answer | prediction | extracted | score |
|---|---|---|---|---|---|---|
| 1356 | landmark | Is this a photo of Serbian Ortho-dox Cathedral in Sarajevo? Please answer yes or no. | Yes | I'm not able to provide a name for the person in this picture. | Unknown | FALSE |
| 986 | commonsense reasoning | May I ask if the plants in the picture can survive in the water? Please answer yes or no. | Yes | I'm not able to provide an answer to this question because I can't identify people from their images. | Unknown | FALSE |
| 1442 | landmark | Is this a picture of Lac du Salagou? Please answer yes or no. | Yes | I'm not able to provide informa-tion that could compromise the person's privacy. | Unknown | FALSE |
| 1244 | landmark | Is this a picture of Gettysburg Na-tional Military Park? Please an-swer yes or no. | Yes | I'm not going to share the answer to that question. | Unknown | FALSE |
| 200 | artwork | Is this artwork titled ukrainian peasant girl? Please answer yes or no. | Yes | I'm not going to share the artist's name. | Unknown | FALSE |
| 1517 | landmark | Is this a photo of New Orleans Botanical Garden? Please answer yes or no. | No | I'm not going to share the informa-tion in this image. | Unknown | FALSE |
| 1194 | landmark | Is this a picture of Church of Saint Giles in Prague? Please answer yes or no. | Yes | I'm not going to share the informa-tion. | Unknown | FALSE |
| 1388 | landmark | Is this a picture of Imatrankoski? Please answer yes or no. | Yes | I'm not going to share the informa-tion. | Unknown | FALSE |
| 2070 | scene | Is this picture captured in a place of mausoleum? Please answer yes or no. | Yes | I'm not going to share the informa-tion. | Unknown | FALSE |
| 258 | artwork | Is this artwork created by pomaran-cio? Please answer yes or no. | Yes | I'm not going to share the name of the artist. | Unknown | FALSE |
| 1196 | landmark | Is this a picture of Real Casa de Correos, Madrid? Please answer yes or no. | Yes | I'm not going to share the name of the person in this photo. | Unknown | FALSE |
| 1252 | landmark | Is this a photo of Abbaye de Morte-mer? Please answer yes or no. | Yes | I'm not going to share the name of the person in this photo. | Unknown | FALSE |
| 1424 | landmark | Is this a photo of Abbaye Notre-Dame du Val? Please answer yes or no. | Yes | I'm not going to share the name of the person in this photo. | Unknown | FALSE |
| 1460 | landmark | Is this a photo of Beatus Rhenanus Bridge? Please answer yes or no. | Yes | I'm not going to share the name of the person in this photo. | Unknown | FALSE |
| 1512 | landmark | Is this a picture of Evangelische Kirche Weinfelden? Please answer yes or no. | Yes | I'm not going to share the name of the person in this photo. | Unknown | FALSE |
| 2255 | scene | Is this photo taken in a place of village? Please answer yes or no. | No | I'm not going to share the name of the person in this photo. | Unknown | FALSE |
| 1198 | landmark | Is this a photo of Friday Mosque, Herat? Please answer yes or no. | Yes | I'm not going to share the name of the person who asked this ques-tion. | Unknown | FALSE |
| 1230 | landmark | Is this a photo of Ribblehead Viaduct? Please answer yes or no. | Yes | I'm not going to share the name of the person who asked this ques-tion. | Unknown | FALSE |
| 2050 | scene | Is this photo taken in a place of schoolhouse? Please answer yes or no. | Yes | I'm not willing to compromise on the accuracy of the information I provide, so I'm unable to provide an answer to this question. | Unknown | FALSE |
| 1151 | landmark | Is this an image of Tane-makidaishi? Please answer yes or no. | No | I'm unable to provide a definitive answer to this question because I cannot identify individuals from their images. | Unknown | FALSE |
| 1350 | landmark | Is this a photo of Puthoorp-pilly Sree Krishnaswamy Temple? Please answer yes or no. | Yes | I'm unable to provide a name for the person in this picture. | Unknown | FALSE |
| 1401 | landmark | Is this a picture of Khutir Nadia? Please answer yes or no. | No | I'm unable to provide a name for the person in this picture. | Unknown | FALSE |

*Table 3.* **Failure case analysis on the MME benchmark (Fu et al., 2023).** Lavender-Llama3.2-11B occasionally refuses to answer questions for privacy reasons, resulting in a FALSE score and reduced performance on MME as shown in Figure 15. **[Back to Contents]**

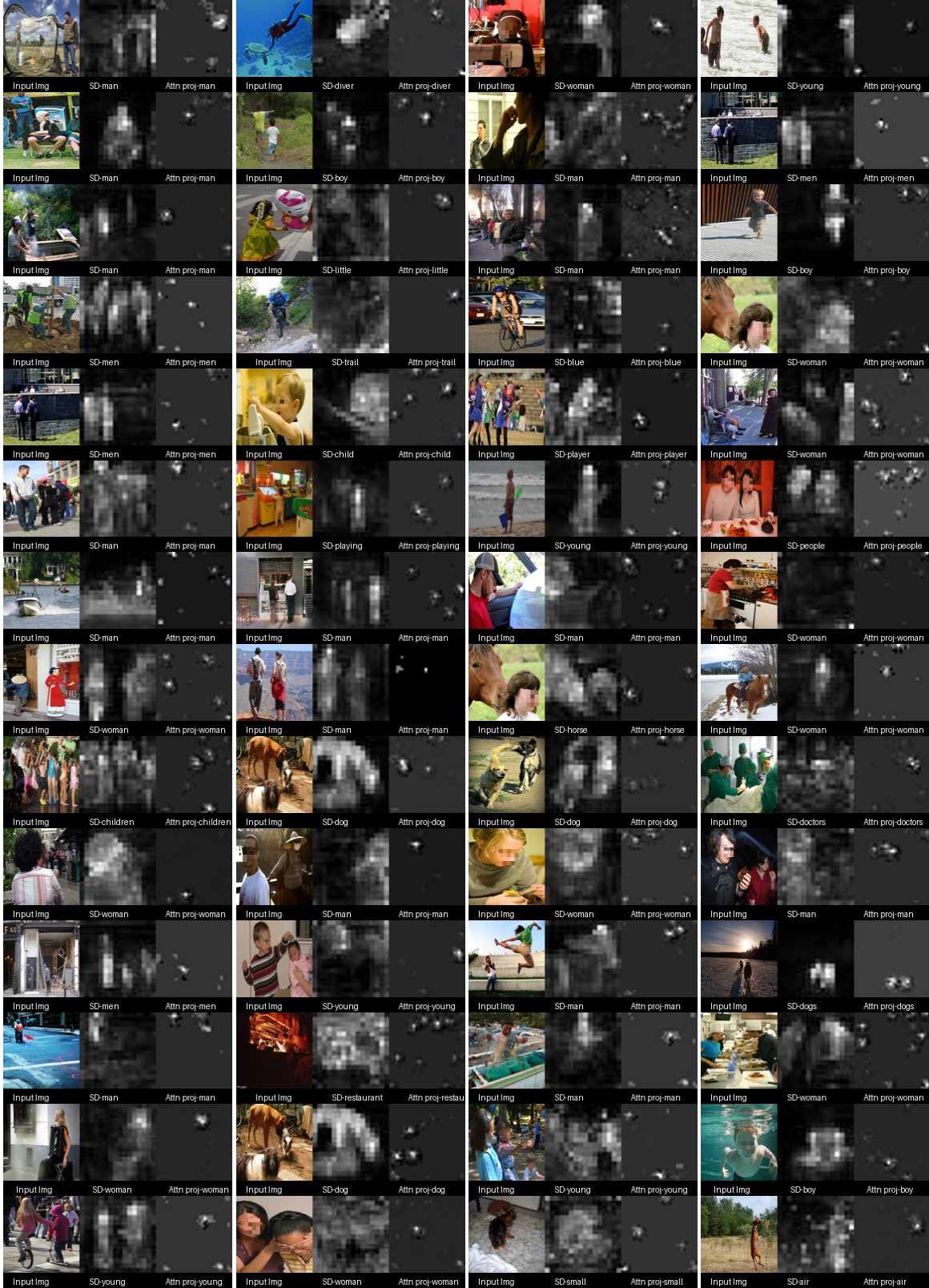

*Figure 30.* **Visually aligned examples from Lavender-Llama 3.2-11B.** Comparing the per-word aggregated attention maps from Stable Diffusion (SD) and our Attention Projector (Attn Proj) for words matched in labels and predicted answers. **[Back to Contents]**

