# OpenReview forum: "Diffusion Instruction Tuning"
_ICML.cc/2025/Conference — ICML 2025 poster_

### Official Review · Reviewer_qVhe · 2025-03-09

**Overall Recommendation:** 4

**Summary:**

This paper introduces Lavender, the first framework designed to directly align the attention layers of vision-language models (VLMs) with Stable Diffusion. Notably, Lavender is model-agnostic, and the authors evaluate it across multiple pretrained VLMs, demonstrating its strong generalization on both in-distribution and out-of-distribution (OOD) data. Furthermore, Lavender fine-tunes efficiently, requiring only 0.13M processed pairs for training.

## update after rebuttal
I have read the authors' responses. The authors have provided useful additional experiments, but not strong enough to justify a 5. Since my original score was 4, I will maintain it.

**Claims And Evidence:**

Yes

**Essential References Not Discussed:**

No

**Experimental Designs Or Analyses:**

Yes, the experimental designs in sections 5, 6, 7 is reasonable and valid.

**Methods And Evaluation Criteria:**

The paper fails to discuss whether the choice of diffusion architecture matters.
- The paper leverages SDv1.4 which uses UNet architecture. Does the analysis hold for DiT-based models?

**Other Comments Or Suggestions:**

Figure 1 could benefit from more distinguishable colors for the baseline to improve readability.

**Other Strengths And Weaknesses:**

Weakness:
- The mathematical symbols used throughout the paper are often not clearly defined.
  - The symbols ($x$, $y_l$, $y_q$, etc.) in sec 2.1 lack explicit definitions.
  - The meaning of $h, l, t, p$ in L216 $w^{hl}_{(t,p)}$ are not specified.
- In Line 1433, it is stated that $A^{(l)} \in R^{N_{text} \times N_{patch}}$, however, the aggregation function used over multi-head is not explicitly described.
- The implementation details for "root word match" and "exact word match" are unclear. Given that text is tokenized into subwords rather than words, how are these matches computed?
- In Figure 9, it appears Lavender shows negative gain in hateful memes, the paper should provide analysis and discussion to explain the result
- In Figure 10, Lavender fails to outperform AR full-FT on OCRBench and DocVQA, both of which involve text recognition. This raises several questions like:
  - Does Lavender not work well in improving the model’s ability to recognize text?
  - Could this be related to the text generation capabilities of diffusion models? Or is it due to the current attention aggregation method?
  - Would switching to a diffusion model with stronger text generation abilities improve performance?
- Additionally, the paper could provide qualitative examples for OCR-related tasks to visualize the attention map in such cases.

**Questions For Authors:**

See above

**Relation To Broader Scientific Literature:**

This work represents an effort to leverage diffusion models (image generation), to enhance image understanding tasks. The proposed method demonstrates robust performance gains, suggesting a promising direction for harnessing the capabilities of generative models to improve understanding tasks.

**Theoretical Claims:**

Some of the assumptions in the Bayesian justification appear to be too strong.
- The assumption that a single ideal attention mechanism exists could be too strong
- In Appendix G, the assumption that the cross-entropy terms are approximately equal seems too strong.

The transition from Equation (23) to (25) is unclear—how does $y_q$ get replaced by $y$?

---

> ### Author Rebuttal · Authors · 2025-04-01
>
> ### **Methods And Evaluation Criteria**:
>
> > "… the choice of diffusion architecture matters … Does the analysis hold for DiT-based models?"
> >
>
> Thank you for highlighting this important point, previously discussed with Reviewers Xyv6 and xQnL. Briefly, Lavender's effectiveness indeed depends on the chosen diffusion model. We conducted additional [visual]((https://anonymous.4open.science/r/2134/1_different_attns/1.jpg)) and [quantitative experiments](https://anonymous.4open.science/r/2134/README.md) using Stable Diffusion v1.4, v2, and Flux (with cross-attention and ConceptAttention [1]); the detailed setup is provided in the table caption. These experiments showed generally improved attention quality and performance with advanced diffusion models, despite persistent OCR-related challenges.
>
> [1] Helbling et al., 2025. ConceptAttention: Diffusion Transformers Learn Highly Interpretable Features.
>
> ---
>
> ### Theoretical Clarifications and Concerns:
>
> > "Some of the assumptions … too strong.
> >
> > - …single ideal attention ...
> > - In Appendix G ...
> > - The transition from …
>
> We thank the reviewer for this detailed and constructive feedback and respond as follows:
>
> - We agree that assuming a single ideal attention mechanism is too strong, as transformer attentions are high-dimensional and task-specific across dimensions. We will clarify that our reference is to ideal attention in vision-centric tasks—Lavender’s focus—and thus motivate our design choice to aggregate high-dimensional attention into single-channel per-word maps to model word-to-region correlations (lines 216–219). Additionally, our Aligner network helps prevent overwriting attentions useful for other tasks, addressing potential interference, as positively noted by Reviewer Xyv6:
>
>     *"Importantly, they introduce measures to handle catastrophic forgetting: e.g., an Aligner network (small learnable layers) to project VLM attention into the diffusion attention space and strategies like LoRA fine-tuning to preserve original model capabilities."*
>
> - Regarding the assumption of approximately equal cross-entropy terms, we clarify that it specifically applies to vision-centric word-to-region attention correlation rather than the entire VLM attention.
> - Concerning the transition from Equation (23) to (25), we clarify:
>     - The VLM processes an image $x$, question $y_q$, and answer label $y_l$, modeling $p(y_l | x, y_q; \theta)$.
>     - The Diffusion Model (DM), however, is conditioned on a unified text $y$, modeling $p(x | y; \theta_D)$.
>     - In our Preprocessing Stage 1 (Algorithm 2), the DM processes image-question pairs, hence replacing $y_q$ with $y$. We will clarify this explicitly in the revision.
>
> ---
>
> ### Weaknesses (clarity  and definitions):
>
> > "The mathematical ...
> >
> > - The symbols ...
> > - The meaning ...
> > - In Line 1433 ...
> > - The implementation details …
>
> We thank the reviewer for identifying these points and acknowledge that clarity was compromised by space constraints. We will revise the manuscript to explicitly define:
>
> - Symbols $x, y_q, y_l$, as image, question, and label answer, respectively.
> - Symbols $h, l, t, p$ in $w_{(t,p)}^{hl}$, indicating a single attention weight where $h \in H, l \in L, t \in T, p \in P$, with $H, L, T, P$ being heads, layers, tokens, and patches, respectively.
> - Clarify the mean or max aggregation function is applied over the multi-head attention.
> - Clarify "root word match" and "exact word match" are post-processing steps on fully generated and decoded answers prior to loss computation and backpropagation.
>
> > "In Figure 9 .."
> >
>
> The negative gain observed in the hateful memes dataset arises primarily because it uniquely employs a ranking classification task, unlike the captioning tasks used for training (Laion50k, Flickr30k) and the other six benchmarks, as specified in lines 1721-1725. We will explicitly add this analysis in the revised Figure 9 discussion.
>
> > "In Figure 10 ..."
> >
>
> Thank you for highlighting this point, previously discussed with Reviewers Xyv6 and 2Jc6. We observed degraded performance when mixing OCRVQA datasets, mainly due to the diffusion model's weaker text attention compared to object recognition. Visual examples illustrating these limitations were provided in earlier responses ([anonymized link](https://anonymous.4open.science/r/2134/2_challenge/1.jpg)).
>
> Since Lavender is model-agnostic, alternative diffusion or OCR-specialized models could enhance performance. Specifically, we suggest:
> - Leverage attention maps from specialized OCR models.
> - Using more advanced diffusion models (e.g., Flux with ConceptAttention [1]).
> - Increasing inversion steps, as shown in preliminary experiments ([anonymized link](https://anonymous.4open.science/r/2134/3_scale_inversion/1.jpg)).
>
> ---
>
> ### Other Comments or Suggestions:
>
> > "Figure 1 could benefit from more distinguishable colors for the baseline to improve readability."
> >
>
> We will fix this in the revised version.

---

> > ### Comment · Reviewer_qVhe · 2025-04-07
> >
> > Thank you for the response and additional experiments! I will maintain the score.

---

### Official Review · Reviewer_xQnL · 2025-03-10

**Overall Recommendation:** 3

**Summary:**

The paper introduces Lavender, a supervised fine-tuning (SFT) method for enhancing vision-language models (VLMs). It aligns the core transformer attention in VLMs with the attention maps of Stable Diffusion during SFT. This approach enriches the model's visual understanding, improves text generation quality, and is highly data-efficient, requiring only 0.13 million training examples. Experiments on multiple VLMs, such as Llama-3.2-11B and MiniCPM-Llama3v2.5, show significant performance gains of up to 30% on various benchmarks and a 68% boost on out-of-distribution tasks like WorldMedQA. The authors also conduct ablation studies to analyze the key components of Lavender.

**Claims And Evidence:**

Claims: The paper claims that Lavender can effectively improve the performance of VLMs, enhance word-to-region alignment, and be more data-efficient than traditional methods.

**Essential References Not Discussed:**

No essential references seem to be missing. The paper covers a wide range of related works, from the development of VLMs and DMs to existing fine-tuning and alignment methods.

**Experimental Designs Or Analyses:**

Experimental Designs: The experimental designs are generally sound. The authors test Lavender on different VLMs with cross-attention and self-attention mechanisms. They vary the training datasets, fine-tuning strategies (e.g., LoRA and full fine-tuning), and attention aggregation methods. The use of a smaller OpenFlamingo model for initial verification and then scaling up to larger models is a logical approach.

**Methods And Evaluation Criteria:**

Yes.

**Other Comments Or Suggestions:**

It would be interesting to see the performance of Lavender on more diverse and larger datasets to better understand its scalability.

Exploring the use of different diffusion models or incorporating additional visual information sources could further enhance the method.

**Other Strengths And Weaknesses:**

Strengths:

Novelty: The idea of aligning VLM attention with that of Stable Diffusion is innovative and shows great potential for improving VLMs.

Detailed formalization: The authors establish a Bayesian framework to formalize the objective of aligning the attention mechanism of Vision-Language Models (VLMs) with that of Diffusion Models (DMs).


Data-efficiency: Requiring only 2.5% of typical large-scale SFT datasets makes Lavender a practical and resource-friendly solution.
Generalizability: The strong performance on out-of-distribution tasks like WorldMedQA demonstrates its potential for real-world applications.



Weaknesses:


Only experiment on Stable Diffusion v1.4: Using an older version of Stable Diffusion may limit the accuracy of attention maps. Upgrading to higher-resolution models could improve performance but also bring resource challenges. I am curious whether experiments with SD XL or Flux or even video diffusion models will bring new phenomena.

**Questions For Authors:**

No.

**Relation To Broader Scientific Literature:**

The paper clearly situates its key contributions in the context of the broader scientific literature. It reviews the development of VLMs, the challenges in training them, and existing approaches to address these challenges. It also discusses how Lavender differs from previous methods, highlighting its novelty in directly aligning VLM transformer attention layers with Stable Diffusion.

**Theoretical Claims:**

The paper provides a Bayesian framework to justify the inclusion of the attention alignment loss. The assumptions made in the framework, such as the DM's attention being closer to the optimal posterior attention distribution, are reasonable and supported by empirical evidence (e.g., lower entropy of DM attention). The derivations are clear and contribute to the theoretical foundation of the Lavender method.

---

> ### Author Rebuttal · Authors · 2025-03-31
>
> ### Weaknesses and Limitations:
>
> > "Only experiment on Stable Diffusion v1.4 …"
> >
>
> Thank you for highlighting this important point. We acknowledge that Lavender’s performance indeed depends on the chosen diffusion model. While our current results with Stable Diffusion v1.4 demonstrate strong attention quality compared to standard VLMs on general images, we recognize its limitations, particularly on tasks like OCR, as previously discussed with Reviewer Xyv6.
>
> To address your concerns and demonstrate Lavender’s generalizability, we've prepared visual examples comparing diffusion models (**Stable Diffusion v1.4**, **Stable Diffusion v2**, and **Flux** with cross-attention and ConceptAttention [1]) across various image types, accessible via [an anonymized link](https://anonymous.4open.science/r/2134/1_different_attns/1.jpg). These examples generally show improved attention quality with more advanced models, though challenges remain, especially in OCR tasks, as illustrated [here](https://anonymous.4open.science/r/2134/2_challenge/1.jpg).
>
> Additionally, we quantitatively evaluated Lavender by extracting attention from Flux (the latest DiT-based model) using cross-attention and ConceptAttention. Due to computational constraints, we limited the evaluation to approximately 2,000 OCRVQA image-text pairs, fine-tuned Lavender-Llama-3.2 with LoRA, and tested across eight benchmarks. Preliminary results (shown below) indicate that better attention from advanced models further improves Lavender’s effectiveness, supporting its model-agnostic capability.
>
> ---
>
> | Attention Model | DocVQA_VAL | InfoVQA_VAL | MME | MMMU   (val) | OCRBench | POPE | RealWorldQA | HallusionBench   (overall) |
> | --- | --- | --- | --- | --- | --- | --- | --- | --- |
> | SD14 Cross Attn | 73.26 | 47.71 | 1721.16 | 39.67 | 707 | **88.49** | 54.12 | **29.09** |
> | Flux Cross Attn | 74.05 | 48.01 | 1706.30 | 39.22 | 716 | 88.43 | 54.51 | 26.97 |
> | Flux Concept Attn | **78.47** | **51.88** | **1787.91** | **39.78** | **750** | 88.20 | **57.26** | 27.67 |
>
> [1] Helbling et al., 2025. ConceptAttention: Diffusion Transformers Learn Highly Interpretable Features.
>
> ---
>
> ### Other Comments or Suggestions:
>
> > "It would be interesting … understand its scalability."
> >
>
> We thank the reviewer for highlighting this important point. Indeed, as explicitly noted in the Limitations and Future Works (lines 405-409) and Appendix D (lines 1140-1141): *"Lavender was evaluated on datasets of up to 0.13M samples, constrained by available compute resources."* This constraint primarily reflects our limited compute resources rather than any inherent scalability issue with Lavender.
>
> Nevertheless, we did examine Lavender’s scaling within our limits (Section 6.4 and Figure 13), showing continued improvement with increasing data size. Appendix D (lines 1142-1143) specifically states this:
>
> *"Figure 17 demonstrates non-convergent scaling behaviour, suggesting that further scaling of both dataset size and tuning length could lead to additional improvements in overall performance with Lavender."*
>
> We hope these findings encourage resource-rich groups to further explore this "*avenue for cross-model knowledge transfer,*" as Reviewer Xyv6 described, and this promising "*direction for harnessing the capabilities of generative models,*" as noted by Reviewer qVhe.
>
> Finally, as emphasized in Section 10 (Impact Statement, lines 463-476), Lavender uniquely benefits smaller research groups, enabling efficient knowledge transfer from large pre-trained models without extensive resources:
>
> *"Data Scarcity. Both the language and vision communities face current or impending data shortages...  End-to-end training from scratch is resource-intensive and often infeasible. Large-scale LLM-finetuned VLMs and DMs have been trained on multi-billion-level datasets, making it inefficient if their knowledge remains isolated. Lavender offers a new approach to bridge these large model families using limited resources—requiring as little as a few thou- sand data points and one day of training on 8 Nvidia A10G GPUs (24GB memory each)—while enabling small models (<13B) to achieve performance on par with large models (>50B) across multiple benchmarks."*
>
> ---
>
> > "Exploring the use of different diffusion models or incorporating additional visual information ..."
> >
>
> We appreciate this insightful recommendation. Indeed, Lavender is fundamentally model-agnostic and is designed to leverage diverse and more specialized attention sources.
>
> Despite early discussed examples, in future work, we envision several directions to explore this further:
>
> - Leverage attention maps from specialized OCR models.
> - Using more advanced diffusion models (e.g., Flux).
> - Increasing inversion steps, as shown in preliminary experiments ([anonymized link](https://anonymous.4open.science/r/2134/3_scale_inversion/1.jpg)).
>
> We will incorporate these insights into the manuscript to clarify future research directions.

---

### Official Review · Reviewer_2Jc6 · 2025-03-13

**Overall Recommendation:** 4

**Summary:**

The paper introduces Lavender, a novel framework that enhances image-to-text generation in vision-language models by aligning their attention mechanisms with text-to-image  diffusion models, specifically Stable Diffusion. The key motivation is that diffusion models, which reconstruct images at the pixel level, capture more precise attention maps with finer spatial granularity than standard VLMs optimized solely for textual output.

**Claims And Evidence:**

The claims made in the submission are largely supported by empirical evidence, including benchmark evaluations, ablation studies, and theoretical justification through Bayesian reasoning.

**Essential References Not Discussed:**

No, I'm not familiar with this area. But I think the author has great references.

**Experimental Designs Or Analyses:**

The experimental design is well-structured, testing Lavender on large scale benchmarks and  baseline models. The study controls for data overlap  to prevent benchmark leakage, ensuring fair evaluation. The paper also examines scaling behavio, showing that Lavender improves generalization without overfitting.

**Methods And Evaluation Criteria:**

The paper introduces Lavender, which aligns VLM attention with that of Stable Diffusion to enhance image-to-text tasks. The benchmarks used, including question answering, captioning, and out-of-distribution tests, appropriately measure the model’s ability to generalize and handle real-world scenarios. The inclusion of WorldMedQA for multilingual medical questions is particularly effective in demonstrating Lavender’s robustness to domain shifts, making the evaluation framework suitable for this application.

**Other Comments Or Suggestions:**

Minor Typos –

"Toekn" → "Token" (multiple occurrences)

"alginment" → "alignment"

**Other Strengths And Weaknesses:**

The experimental evaluation dataset in the paper has a maximum of only 0.13M samples, which is much smaller than the 5M to 50M level datasets used in existing technology models.This limits the ability to fully evaluate Lavender's scalability with larger data sizes.

Mixing OCRVQA datasets with other datasets can sometimes degrade performance, implying a risk of overfitting on specific data.

**Questions For Authors:**

The paper shows strong quantitative results, but it does not discuss where Lavender fails or underperforms. Could authors provide a deeper failure case analysis? Are there scenarios where Lavender struggles, such as highly complex or ambiguous image-text relationships?

**Relation To Broader Scientific Literature:**

The paper builds on vision-language models fine-tuning and diffusion model attention mechanisms. Prior works aligned image-text representations at the encoder level, but Lavender is the first to align transformer attention maps directly. It extends ideas from Stable Diffusion cross-attention and VLM tuning (LLaVA, OpenFlamingo), improving word-to-region alignment. Compared to autoregressive fine-tuning, Lavender shows better generalization with minimal data. Its focus on OOD robustness connects to multimodal domain adaptation, expanding prior research in efficient multimodal learning while reducing data reliance.

**Theoretical Claims:**

The proofs are logically structured but rely on empirical justification rather than formal theoretical guarantees.

---

> ### Author Rebuttal · Authors · 2025-03-31
>
> ### Weaknesses:
>
> > "The experimental evaluation dataset ... scalability with larger data sizes."
> >
>
> We thank the reviewer for highlighting this important point. Indeed, as explicitly noted in the Limitations and Future Works (lines 405-409) and Appendix D (lines 1140-1141): *"Lavender was evaluated on datasets of up to 0.13M samples, constrained by available compute resources."* This constraint primarily reflects our limited compute resources rather than any inherent scalability issue with Lavender.
>
> Nevertheless, we did examine Lavender’s scaling within our limits (Section 6.4 and Figure 13), showing continued improvement with increasing data size. Appendix D (lines 1142-1143) specifically states this:
>
> *"Figure 17 demonstrates non-convergent scaling behaviour, suggesting that further scaling of both dataset size and tuning length could lead to additional improvements in overall performance with Lavender."*
>
> We hope these findings encourage resource-rich groups to further explore this "*avenue for cross-model knowledge transfer,*" as Reviewer Xyv6 described, and this promising "*direction for harnessing the capabilities of generative models,*" as noted by Reviewer qVhe.
>
> Finally, as emphasized in Section 10 (Impact Statement, lines 463-476), Lavender uniquely benefits smaller research groups, enabling efficient knowledge transfer from large pre-trained models without extensive resources:
>
> *"Data Scarcity ...  End-to-end training from scratch is resource-intensive and often infeasible. Large-scale LLM-finetuned VLMs and DMs have been trained on multi-billion-level datasets, making it inefficient if their knowledge remains isolated. Lavender offers a new approach to bridge these large model families using limited resources—requiring as little as a few thousand data points and one day of training on 8 Nvidia A10G GPUs (24GB memory each)—while enabling small models (<13B) to achieve performance on par with large models (>50B) across multiple benchmarks."*
>
> ---
>
> > "Mixing OCRVQA datasets ..."
> >
>
> Indeed, we observed performance degradation when mixing OCRVQA datasets with others, primarily due to the limited OCR capabilities of the employed diffusion model. Diffusion models, optimized mainly for image generation, tend to exhibit weaker attention on text compared to general object recognition. We previously discussed this limitation extensively with Reviewer Xyv6, providing visual evidence through [an anonymized link](https://anonymous.4open.science/r/2134/2_challenge/1.jpg) to illustrate attention map failures, specifically for OCR tasks.
>
> Lavender is model-agnostic and can leverage more appropriate teacher models with better-aligned attention maps, as noted in Section 10, Impact Statement, page 9, line 494:
>
> *"Currently, Lavender’s alignment objectives are derived from Stable Diffusion model’s attention maps. However, the same approach could be applied to other vision foundation models with well-aligned per-word attention maps."*
>
> Therefore, to address this limitation, we propose the following potential solutions:
>
> - Leverage attention maps from specialized OCR models.
> - Using more advanced diffusion models (e.g., Flux with ConceptAttention [1]), with [visual](https://anonymous.4open.science/r/2134/1_different_attns/1.jpg) and [quantitative results](https://anonymous.4open.science/r/2134/README.md).
> - Increasing inversion steps, as shown in preliminary experiments ([anonymized link](https://anonymous.4open.science/r/2134/3_scale_inversion/1.jpg)).
>
> [1] Helbling et al., 2025. ConceptAttention: Diffusion Transformers Learn Highly Interpretable Features.
>
> ---
>
> ### Questions:
>
> > "The paper shows strong quantitative results, but it does not discuss where Lavender fails or underperforms. ..."
> >
>
> We thank the reviewer for this insightful question. We summarize previously discussed and newly identified failure cases below:
>
> 1. We explicitly discussed unsuccessful training strategies for Lavender in the Failure Strategies section (page 8, line 429) and Appendix C.
> 2. Additionally, as discussed earlier, one key scenario where Lavender underperforms is when the "teacher" diffusion model does not provide sufficiently high-quality attention, such as in OCR tasks. Examples can be found [here](https://anonymous.4open.science/r/2134/2_challenge/1.jpg). These results underscore the diffusion model’s specific limitations in accurately attending to highly complex and ambiguous textual information, negatively impacting Lavender’s OCR-VQA performance.
>
> We will further expand our manuscript to explicitly discuss Lavender’s performance limitations in highly complex or ambiguous scenarios beyond OCR, including instances of subtle semantic distinctions or particularly abstract image-text relationships.
>
> ---
>
> ### Minor Typos:
>
> > "Toekn" → "Token" (multiple occurrences), "alginment" → "alignment"
> >
>
> We thank the reviewer for highlighting these and will correct them in our revised version.

---

### Official Review · Reviewer_Xyv6 · 2025-03-14

**Overall Recommendation:** 3

**Summary:**

Diffusion Instruction Tuning introduces Lavender, a fine-tuning framework that aligns a vision-language model’s (VLM) image-to-text attention with a text-to-image diffusion model’s attention maps​. The key idea is to leverage the precise cross-attention of a pretrained Stable Diffusion model as a training signal for the VLM: during supervised fine-tuning on image-text pairs, Lavender adds an auxiliary loss that pushes the VLM’s token-level attention to mimic the diffusion model’s attention, alongside the usual next-token prediction loss​. This is the first approach to directly align transformer attention layers of a VLM with a diffusion model’s attention (prior works only aligned at image encoder or adapter levels)

**Claims And Evidence:**

- Diffusion models have more precise attention maps than VLMs.

The authors claim that a text-to-image diffusion model (Stable Diffusion) learns fine-grained word-to-region alignment that VLMs lack. This is backed by qualitative and quantitative evidence: Figure 2 compares attention maps, showing diffusion’s per-word attention is more tightly focused on relevant image regions than a VLM’s​. They also report significantly lower entropy in diffusion attention distributions, indicating they are more peaked and informative​. This empirical evidence supports the premise that diffusion attention is a good target for alignment.

- Aligning VLM attention to diffusion attention improves performance.

This is the core hypothesis, and it is strongly validated by extensive experiments. Lavender fine-tuning consistently outperforms standard instruction tuning (next-token loss only) across diverse tasks. For instance, on an OpenFlamingo model, Lavender yields up to +70% relative improvement across several benchmarks​. On a larger Llama-3.2 11B model, Lavender improves accuracy by up to 30% on 19/20 benchmarks compared to autoregressive fine-tuning, and even surpasses comparable open-source models by ~50%​. Even a self-attention-only model (MiniCPM) sees gains (up to 4%)​. These results provide convincing evidence that the alignment loss delivers tangible benefits.

**Essential References Not Discussed:**

The related work coverage is extensive.

**Experimental Designs Or Analyses:**

The experimental design is a strong point of the paper, with thorough evaluations and thoughtful analyses. The authors trained and tested on a wide variety of datasets, which reduces the chance of overfitting the method to a particular benchmark. They compiled ~130k image-text pairs for fine-tuning, drawn from multiple sources (referred to as RV83k, Flk30k, OV30k in the paper) – this mix includes general and possibly task-specific data, ensuring the model is exposed to diverse scenarios​. While this is a relatively small corpus, it was intentional to demonstrate data-efficiency; it also means the base models are not pushed to their limit capacity, highlighting the effect of the alignment rather than brute-force data. On the evaluation side, the use of 20 benchmarks covering four broad categories (charts/docs, perception, real-world, and no-hallucination tasks) is appropriate to claim broad generalization​. The results are reported with clear metrics (often using zero-shot evaluation on each benchmark), and they even provide relative improvements which make it easy to see the benefit of Lavender over baselines.

**Methods And Evaluation Criteria:**

The proposed method is well-described and appropriate for the problem. Lavender introduces a two-stage fine-tuning procedure: first, precompute the diffusion model’s cross-attention maps for each training image-text pair; second, fine-tune the VLM with a combined loss (standard language modeling loss plus an attention alignment loss)​. This approach directly addresses the stated goal of improving visual-text alignment, by explicitly training the model to align its internal attention with a more grounded reference. The method is implemented in a model-agnostic way – the loss can be applied to any VLM, with either explicit cross-attention layers or even unified self-attention (the authors devise an attention aggregation for the latter case). They provide clear pseudocode (Algorithm 1) and discuss how to aggregate attention across heads/layers in both diffusion models and VLMs​. Importantly, they introduce measures to handle catastrophic forgetting: e.g. an Aligner network (small learnable layers) to project VLM attention into the diffusion attention space​ and strategies like LoRA fine-tuning to preserve original model capabilities​. These design choices seem appropriate – aligning at the transformer attention is a sensible place (it’s the “core” of vision-language interaction), and the use of a mean-squared error loss on attention distributions is a natural choice for guiding one distribution toward another​. The method is novel yet grounded in known techniques (it can be seen as a form of knowledge distillation on attention maps), and it’s evaluated thoroughly.

**Other Comments Or Suggestions:**

The result on MiniCPM (which lacks a dedicated cross-attention module) showed smaller gains (~4%). It would be useful to add a bit more discussion on why the improvement was limited there – is it because the model’s architecture makes it harder to inject the alignment (since image and text tokens attend to each other in the same layers)? The authors hinted that cross-attention models improved more strongly than self-attention models​. This is a valuable insight: it suggests that having an explicit cross-attention makes alignment easier to enforce. Perhaps a brief mention in the paper (if space permits) to explain this difference would be enlightening to readers considering Lavender for different model types.

**Other Strengths And Weaknesses:**

Strengths:

-  The idea of using a diffusion model’s attention maps as supervision for a VLM is highly original. This is the first work to my knowledge that connects a generative vision model to a descriptive vision-language model in this way​. It opens up a novel avenue for cross-model knowledge transfer.

Weaknesses:

- Dependence on Diffusion Model Quality: Lavender’s success hinges on the teacher model (Stable Diffusion) having good attention maps. If the diffusion model’s attention is poor for certain types of images or concepts, the benefit to the VLM could be limited or even harmful. We saw an example with OCR – Stable Diffusion likely doesn’t attend well to fine text, and indeed adding an OCR task didn’t help​. So a weakness is that the approach inherits the biases/limitations of the diffusion model. If an image is very different from what Stable Diffusion was trained on (e.g., specialized medical imagery or abstract diagrams), its attention might not be “optimal,” potentially limiting Lavender’s performance there. The paper highlights the positive side (it still helped on medical QA), but it’s conceivable there are scenarios where this attention transfer provides little gain or requires a different teacher.

**Questions For Authors:**

- Diffusion Attention Extraction:

Could the authors elaborate on how you obtain the diffusion model’s attention maps for a given image x and text y? Specifically, since Stable Diffusion is a text-to-image model, do you perform some form of image reconstruction or noise conditioning with the real image to get its attention? (For example, do you encode the image into the latent space and run the diffusion model’s denoising steps while conditioning on y to collect cross-attention?) Clarifying this process would help readers understand how $p_{DM}(a|x,y)$ is computed in practice.

- Handling of OCR/Text in Images

The authors observed that including an OCR-VQA dataset led to worse performance​ presumably because the diffusion model doesn’t handle text in images well. How might one address this? Do you think using a different teacher model for text regions (such as an OCR model’s attention or a captioning model trained for text) could be combined with Lavender? Or would you suggest simply excluding or separately treating tasks that involve reading text from images? It would be insightful to hear your thoughts on extending the method to handle scenarios where the diffusion model’s attention might not be reliable.

**Relation To Broader Scientific Literature:**

This work sits at the intersection of visual instruction tuning and diffusion models for vision, offering a novel cross-over between the two areas. In the context of vision-language models (VLMs), recent progress like Flamingo and OpenFlamingo introduced architectures with cross-attention to connect image and text features​, and methods like LLaVA (Visual Instruction Tuning by Liu et al., 2023) showed that one can take a pretrained language model and fine-tune it with a relatively small set of image-text examples (on the order of 150k) to endow it with multimodal capabilities​. Lavender builds on this line of work by addressing a specific weakness of such instruction-tuned models: their visual grounding is often coarse or suboptimal because the training primarily optimizes text outputs. Prior approaches to mitigate this include using adapter modules or refined image encoders. For example, LLaVA and similar methods attach an MLP or a projection module to feed visual features into the LLM and fine-tune that on QA pairs​. Other works have tried to improve visual grounding by enhancing the image encoder or using multiple vision experts – e.g., CoCa and BLIP-2 align encoders with the language model, and some 2024 works merge features from several specialist models via learned projections​. However, all these approaches still operate at the feature level (aligning outputs of encoders or adding new layers) rather than aligning the internal attention behavior of the model.

**Theoretical Claims:**

No obvious correctness issues are found in the theoretical development – it’s a straightforward application of Bayesian thinking to justify an auxiliary loss. The mathematical steps (detailed in Appendix G/H) fill in the gaps for interested readers. One could argue that the assumption that Stable Diffusion’s attention is nearly optimal might not hold in all cases (it’s possible the diffusion model attends to certain aspects that are useful for generation but not for understanding, or misses some semantics). The authors acknowledge this is an approximation, supported heuristically by the entropy observations rather than a formal proof. However, given the difficulty of defining a “ground-truth” attention distribution, the argument is plausible and consistent with the results. In summary, the theoretical claims are well-aligned with the empirical approach. The use of Bayesian terminology lends a principled perspective, and while it doesn’t rigorously prove that this is the optimal training strategy, it provides a solid rationale for why aligning to diffusion’s attention should help (i.e., it injects an informative prior for the VLM’s attention). There were no apparent mathematical mistakes in the derivations provided. The paper could improve by discussing any conditions where the assumption might break (e.g., if the diffusion model’s training data is very different from the VLM’s domain), but overall the theoretical component is a correct and helpful explanation of the method’s foundation.

---

> ### Author Rebuttal · Authors · 2025-03-31
>
> ### Weaknesses:
>
> > "Dependence on Diffusion Model Quality ..."
> >
>
> Thank you for highlighting this important point. We acknowledge that Lavender’s effectiveness indeed depends on the quality and training domain of the chosen diffusion model. While our qualitative and quantitative results confirm that Stable Diffusion v1.4 provides superior attention compared to VLMs on general images, its performance on tasks such as OCR can be limited.
>
> We wish to emphasize that Lavender is fundamentally model-agnostic and is not constrained to a particular diffusion or VLM model, as noted in our submission (Section 10, Impact Statement, p.9, line 494):
>
> *"... the same approach could be applied to other vision foundation models with well-aligned per-word attention maps."*
>
> To address your concerns and demonstrate Lavender’s generalizability, we've prepared visual examples comparing diffusion models (**Stable Diffusion v1.4**, **Stable Diffusion v2**, and **Flux** with cross-attention and ConceptAttention [1]) across various image types, accessible via [an anonymized link](https://anonymous.4open.science/r/2134/1_different_attns/1.jpg). These examples generally show improved attention quality with more advanced models, though challenges remain, especially in OCR tasks, as illustrated [here](https://anonymous.4open.science/r/2134/2_challenge/1.jpg).
>
> Additionally, we quantitatively evaluated Lavender by extracting attention from Flux (the recent DiT-based model) using cross-attention and ConceptAttention. Due to computational constraints, we limited the evaluation to approximately 2,000 OCRVQA image-text pairs, fine-tuned Lavender-Llama-3.2 with LoRA, and tested across eight benchmarks. Preliminary results (shown below) indicate that better attention from advanced models further improves Lavender’s effectiveness, supporting its model-agnostic capability.
>
> ---
>
> | Attention Model | DocVQA_VAL | InfoVQA_VAL | MME | MMMU   (val) | OCRBench | POPE | RealWorldQA | HallusionBench   (overall) |
> | --- | --- | --- | --- | --- | --- | --- | --- | --- |
> | SD14 Cross Attn | 73.26 | 47.71 | 1721.16 | 39.67 | 707 | **88.49** | 54.12 | **29.09** |
> | Flux Cross Attn | 74.05 | 48.01 | 1706.30 | 39.22 | 716 | 88.43 | 54.51 | 26.97 |
> | Flux Concept Attn | **78.47** | **51.88** | **1787.91** | **39.78** | **750** | 88.20 | **57.26** | 27.67 |
>
> [1] Helbling et al., 2025. ConceptAttention: Diffusion Transformers Learn Highly Interpretable Features.
>
> ---
>
> ### Other Comments or Suggestions:
>
> > "The result on MiniCPM ..."
> >
>
> Thank you for pointing this out. Indeed, our experiments suggest that explicit cross-attention modules align more effectively than pure self-attention models like MiniCPM. Dedicated cross-attention modules have parameters specifically for vision-text alignment, whereas self-attention models share parameters across multiple tasks, complicating simultaneous optimization. This distinction resembles how specialized brain regions handle visual and linguistic integration.
>
> We will incorporate a concise version of this explanation into the manuscript.
>
> ---
>
> ### Questions for Authors:
>
> > Diffusion Attention Extraction ..."
> >
>
> Thank you for highlighting this for clarification. We initially summarized this briefly in the appendix (Appendix L, lines 1589-1594):
>
> *" ... We apply a shortened image inversion process (Mokady et al., 2022; Jin et al., 2023) to approximate the text prompt embeddings for image reconstruction, collecting attention maps at each step as in Section 3.1 ..."*
>
> To elaborate further, given an image $x$ and corresponding text $y$, we reconstruct image $x$ by conditioning on the text embedding of $y$, derived using Stable Diffusion’s text encoder. The Null-text inversion (Mokady et al., 2022) further refines this by making the text embedding y learnable, typically requiring extensive inversion steps (e.g., 1000). However, we observed that Stable Diffusion v1.4 already effectively recognizes common concepts from general datasets, enabling us to obtain sufficiently accurate attention maps with just 10 inversion steps.
>
> ---
>
> > Handling of OCR/Text in Images ..."
> >
>
> This is a valuable point, and we propose several potential solutions:
>
> - Leverage attention maps from specialized OCR models.
> - Using more advanced diffusion models (e.g., Flux).
> - Increasing inversion steps, as shown in preliminary experiments ([anonymized link](https://anonymous.4open.science/r/2134/3_scale_inversion/1.jpg)).
> - While separate or integrated architectures may perform similarly for specific OCR tasks, integrated architectures offer the benefit of multitask training, potentially improving general LLM capabilities.
> - For OOD scenarios, scaling inversion steps (similar to test-time compute scaling) is viable if data is limited but computational resources are sufficient. Alternatively, with sufficient data, fine-tuning the diffusion model directly using techniques like LoRA can significantly boost attention quality.

---

### Decision · Program_Chairs · 2025-05-01

**Decision:**

Accept (poster)

**Comment:**

The paper introduces Lavender, a novel framework that enhances image-to-text generation in vision-language models by aligning their attention mechanisms with text-to-image diffusion models, specifically Stable Diffusion. The key motivation is that diffusion models, which reconstruct images at the pixel level, capture more precise attention maps with finer spatial granularity than standard VLMs optimized solely for textual output.

The paper receives 4 positive comments after rebuttal. The claim and evidence are solid and well demonstrated by experiments.